# HESS Opinions: The complementary merits of competing modelling philosophies in hydrology

Markus Hrachowitz[1], Martyn Clark[2]

[1] Faculty of Civil Engineering and Geosciences, Delft University of Technology, Stevinweg 1, 2628 CN Delft, Netherlands
[2] National Center for Atmospheric Research, Boulder CO, 80301, USA

*Correspondence to*: Markus Hrachowitz (m.hrachowitz@tudelft.nl)

**Abstract.** In hydrology, two somewhat competing philosophies form the basis of most process-based models. At one endpoint of this continuum are detailed, high-resolution descriptions of small-scale processes that are numerically integrated to larger scales (e.g. catchments). At the other endpoint of the continuum are spatially lumped representations of the system that express the hydrological response via, in the extreme case, a single linear transfer function. Many other models, developed starting from these two contrasting endpoints, plot along this continuum with different degrees of spatial resolutions and process complexities. A better understanding of the respective basis as well as the respective shortcomings of different modelling philosophies has potential to improve our models. In this manuscript we analyse several frequently communicated beliefs and assumptions to identify, discuss and emphasize the functional similarity of the seemingly competing modelling philosophies. We argue that deficiencies in model applications largely do not depend on the modelling philosophy, although some models may be more suitable for specific applications than others and vice versa, but rather on the way a model is implemented. Based on the premises that any model can be implemented at any desired degree of detail and that any type of model remains to some degree conceptual we argue that a convergence of modelling strategies may hold some value for advancing the development of hydrological models.

## 1 What is the issue?

Hydrological models are used to predict floods, droughts, groundwater recharge and land-atmosphere exchange, and are of critical importance as tools to develop strategies for water resources planning and management. This is particularly true in the light of the increasing effects of climate and land-use change on the terrestrial water cycle. Yet, in spite of their central importance, these models are frequently plagued by considerable uncertainties and unreliable predictions.

Models aim to encapsulate our understanding of the system. Yet, their weakness for predictions suggests that, besides the impact of observational uncertainties, at least some of the processes that control how water and energy are stored in, transferred through and released from different parts of a flow system are not sufficiently well represented in state-of-the-art models.

The hydrologic modelling community sets out to design system descriptions that are explicitly based on our understanding of the actual mechanisms involved. This is done with a wide range of strategies along a two-dimensional continuum of different spatial resolutions and process complexities (Figure 1). Note that hereafter we refer to process complexity as the number of processes that are represented explicitly. At one endpoint of this continuum are detailed, high resolution

5    descriptions of small-scale processes that are numerically integrated to larger scales (e.g. catchments). At the other endpoint of the continuum are spatially lumped representations of the system that express the hydrological response via, in the extreme case, a single linear transfer function. Many other models, developed starting from these two contrasting endpoints, plot along this continuum with different degrees of spatial resolutions and process complexities. Models are then often loosely and informally categorized into these two model classes whose origin roughly reflect the endpoints of the resolution-

10    complexity continuum.

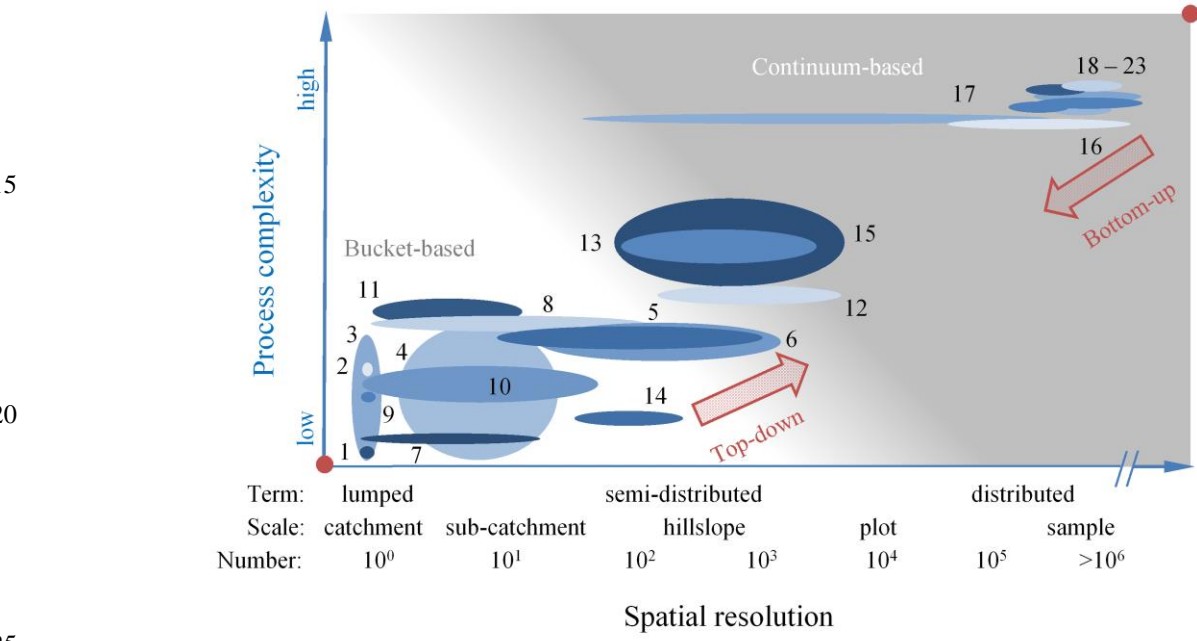

Figure 1: Conceptual sketch of approximate positions of a selection of typical applications of frequently used catchment-scale models on the spatial resolution – process complexity continuum, compared to a selection of observation variables and their availability to force and constrain models. The Spatial resolution axis shows approximate ranges of numbers and associated scales of individual spatial units (e.g. grid cells) within the model domain (e.g. catchment) for typical

30    applications of the individual models. The Process complexity axis indicates the number of individual processes/processes within one spatial unit. The increasingly grey shaded area indicates the transition from bucket-based (white) to continuum-based models. The red dots indicate the two endpoints along the resolution-complexity continuum. Models: 1 – Unit hydrograph (Sherman, 1932), 2 – HBV (Bergström, 1992), 3 – SUPERFLEX (Fenicia et al., 2011), 4 – FLEX-Topo (Gharari et al., 2014a), 5 – mhM (Samaniego et al., 2010), 6 – mhM-topo (Nijzink et al., 2016a), 7 – SWAT (Arnold et al., 1998), 8 –

*NWS-Sacramento (Burnash, 1995), 9 – GR4J (Perrin et al., 2003), 10 – HYPE (Lindström et al., 2010), 11 – VIC (Liang et al., 1994), 12 – TOPMODEL (Beven and Kirkby, 1979), 13 – CRHM, 14 – TAC$^D$ (Uhlenbrook et al., 2004), 15 – WASIM-ETH (Schulla and Kasper, 1998), 16 – DHSVM (Wigmosta et al., 1994), 17 – MIKE-SHE (Refsgaard and Storm, 1996), 18 – PARFLOW (Kollet and Maxwell, 2008), 19 – CATFLOW (Zehe et al., 2001), 20 – HYDRUS-3D (Šimůnek et al., 2008), 21 – CATHY (Camporese et al., 2010), 22 – HydroGeoSphere (Jones et al., 2006), 23 – PIHM (Qu and Duffy, 2007).*

Over the past four decades innumerable studies illustrated the value but also the limitations of models at different positions in the resolution-complexity continuum (Clark et al., 2017). Irrespective of their resolutions and complexities, models can exhibit considerable skill to reproduce the system response dynamics they have been trained for. In spite of that, these models can frequently not simultaneously reproduce aspects of the observed system response other than the calibration objectives, and which may include descriptors of emergent patterns, i.e. catchment signatures, such as flow duration curves (e.g. Jothityangkoon et al. 2001; Eder et al., 2003; Yadav et al., 2007; Martinez and Gupta, 2011; Sawicz et al., 2011; Euser et al., 2013; Willems et al., 2014; Shafii and Tolson, 2015; Westerberg and McMillan, 2015) but also temporal dynamics and/or spatial pattern in state and flux variables the model may not have been calibrated to, such as snow cover (e.g. Parajka and Blöschl, 2008), ground- (e.g. Fenicia et al., 2008) or soil water fluctuations (e.g. Sutanudjaja et al., 2014). This failure to mimic system internal dynamics and patterns in a meaningful way indicates that, while doing a good curve-fitting job, many models may not represent the dominant processes of the system in a meaningful way, thereby providing the right answers for the wrong reasons (cf. Kirchner, 2006). Together with the largely inevitable errors introduced by data uncertainty (e.g. Beven and Westerberg, 2011; Beven et al., 2011; Renard et al., 2011; Beven, 2013; McMillan et al., 2012; Kauffeldt et al., 2013; McMillan and Westerberg, 2015; Coxon et al., 2015) and insufficient model evaluation and testing (cf. Klemes, 1986; Wagener 2003; Clark et al., 2008; Gupta et al., 2008, 2012; Andreassian et al., 2009), models then often experience substantial performance decreases when used to predict the hydrological response for time periods they were not calibrated for (e.g. Seibert, 2003; Refsgaard and Henriksen, 2004; Kirchner, 2006; Coron et al., 2012; Gharari et al., 2013).

Notwithstanding similar skills and limitations of many models along the resolution-complexity continuum, as illustrated by a range of model inter-comparison studies (e.g. Reed et al. 2004; Breuer et al., 2009; Smith et al., 2012; Lobligeois et al., 2013, Maxwell et al., 2014; Vansteeenkiste et al., 2014), there is surprisingly little fruitful exchange between the different modelling communities who start their model development from the two contrasting endpoints in the resolution-complexity continuum. Models at the low resolution and low complexity end of the continuum are criticized for lacking a robust physical or theoretical basis and for their inability to meaningfully represent spatial patterns (e.g. Paniconi and Putti, 2015; Fatichi et al., 2016), whereas models at the high resolution and high complexity end are often viewed as having inferior representations of sub-grid variability (e.g. Beven and Cloke, 2012) and as being not sufficiently agile to represent the dominant processes in different environments (e.g. Mendoza et al., 2015). Even more, instead of appreciating the potential value of a convergence between the approaches and joining forces to integrate the respective efforts, communication

between the communities is often limited to mutually highlighting the deficiencies of and dismissing the respective modelling strategies.

Building on early landmark papers that outline most of the problems involved (e.g. Dooge, 1986; Beven, 1995; Blöschl and Sivapalan, 1995; Beven, 2001; Blöschl, 2001), we think that to achieve progress in the discipline of scientific hydrology and to develop models for more reliable predictions, it is necessary for the different hydrologic modelling communities to take a step back. Reflecting on failures and successes can not only help to design better models but also to better appreciate the *complementary* nature and value of detailed, microscale process understanding on the one hand and the quest for general laws at the macroscale on the other hand (Klemes, 1983; Dooge, 1986; Sivapalan, 2005).

This commentary is based on detailed and, at times, refreshingly heated discussions during and after the *1st Workshop on Improving the Theoretical Underpinnings of Hydrologic Models* (Bertinoro, April 2016). Our aim is to identify, discuss and clarify common misunderstandings and misinterpretations of competing modelling approaches, with the main points being that (1) all models are, to varying degrees, spatially lumped, (2) all models contain, to varying degrees, conceptual elements, (3) all models do have, if well implemented, a sound physical basis, albeit on different scales and (4) the choice of a suitable modelling strategy depends on the purpose of the application. More generally, we intend to resolve the perceived dichotomy between the two modelling communities and their modelling strategies. As many individual points addressed hereafter may have already been discussed elsewhere in more detail we do not make a particular claim for originality. Rather, we want to provide a synthesis of these points with a subsequent perspective of how to take advantage of different modelling philosophies and how convergence between them may be key towards improving both, our understanding of the hydrological system and our hydrological predictions.

**2 Model taxonomy**

Hydrology models are generally classified following a quite loose and informal framework. Models at the low-resolution, low-complexity end of the continuum (Figure 1) are usually referred to as lumped, conceptual, bucket or top-down models. In contrast, high-resolution, high-complexity models are referred to as distributed, physically-based, process or bottom-up models. In spite of having specific meanings and only partial overlap, these individual terms for each modelling strategy are commonly used interchangeably. This lack of a clear and unambiguous terminology may be one of the reasons for many misunderstandings between the different model communities. We therefore think that a somewhat more rigorous model taxonomy needs to be the first step to clarify these misunderstandings and to pave the way for increased convergence of the individual modelling strategies.

The most common model classifications are based on (1) spatial simplification: spatially distributed and spatially lumped; (2) system simplification: physically based and conceptual, (3) model architecture: continuum based and bucket based; and (4) model refinement/scaling strategies: top-down and bottom-up. The following sections describe each of these distinctions in detail.

## 2.1 Spatial simplification: Spatially distributed vs. spatially lumped

### 2.1.1 Spatially distributed models

Spatially distributed models provide, to varying degrees, spatially explicit representations of natural heterogeneity within the model domain. This can be achieved in three ways (e.g. Ajami et al., 2004; Das et al., 2008; Euser et al., 2015): (1) spatially distributed moisture accounting, i.e. each parallel model unit is represented by the same model parameter values but forced with spatially varying model input (e.g. precipitation, temperature, etc.), (2) spatially distributed model parameters that account for heterogeneity in the natural boundary conditions and (3) a combination thereof.

These models can then be further distinguished into two broad functional classes, as suggested by Todini (1988). The first class being a suite of one-dimensional column elements, with no representation of direct lateral exchange between the individual elements ("distributed integral models"). Rather, the elements are merely connected by the channel network. The second class explicitly accounts for lateral exchange of water, solutes and energy between the individual columns ("distributed differential models").

In general, the term "spatially distributed" has limited discriminatory power, as it always needs to be seen with respect to the scale of a specific model application. In addition, the term only describes the spatial axis in the resolution-complexity continuum (Figure 1). However, it is possible to have many different types of spatially distributed models with different degrees of process complexity, model architecture, and model refinement/scaling strategies. In contrast to what the terms imply, different applications of fully distributed models span several magnitudes of grid sizes from centimetres to kilometres (e.g. Butts et al., 2004; Kollet and Maxwell, 2006; Zehe et al., 2006; Samaniego et al., 2010; Kumar et al., 2013) and do thus not necessarily describe the system at a higher spatial resolution than so-called semi-distributed models, as the applied grid cells can often be larger than sub-catchments and/or hydrological response units (e.g. Nijzink et al., 2016a).

It is in any case important to realize that any distributed model application, irrespective of the spatial resolution and scale of its individual model units, is an assemblage of lumped representations of the system at the scale of the individual model units (Grayson and Blöschl, 2001; Wagener and Gupta, 2005).

### 2.1.2 Spatially lumped models

Lumped models represent the model domain as one single entity without further spatial discretization. They describe the large-scale manifestation of small-scale natural heterogeneities of the system by making use of parsimonious flux parameterizations that emerge as functional relationships at the scale of the model domain. Lumped models can be used for systems over a wide range of scales, from soil sample to river basin scale, as long as the emergent relationships meaningfully capture the effects of intra-domain heterogeneity.

## 2.2 System simplification: physically based and conceptual

### 2.2.1 Physically based models

Physically based models provide a mechanistic description of the flow system in the porous and heterogeneous soil column and at the land-vegetation-atmosphere interface, consistent with our understanding of the forces acting on and controlling the release of water, energy and solutes from the control volumes under consideration. They attempt to do so by explicitly representing as many processes as possible (Figure 1). An ideal situation is where there is comprehensive knowledge of boundary conditions (e.g. effective soil hydraulic conductivity, precipitation), system states (e.g. volumetric water content) and fluxes (e.g. canopy throughfall, infiltration, subsurface lateral flow), and it is possible to define functional relationships between states and fluxes (i.e. flux parameterizations), such as storage-discharge relationships in the form of Q=f(S). Such flux parameterizations, or closure relations, then directly emerge at the scale of the observation, fully satisfying the conservation laws of mass, energy and momentum and, in theory, without the need for further assumptions or calibration. As most direct observations of system boundary conditions and states are only available at the point or plot scale, models that are traditionally referred to as physically based may also be considered as describing the system from microscale perspective.

There is an important distinction here. Individual observations provide lumped characterizations of a flow system, integrating spatial heterogeneities and diversity in processes at scales smaller than the scale of the observation (Grayson and Blöschl, 2001). To be meaningful, so-called physically based models are actually lumped at the scale of the observation, as any further discretization below the scale of the observation needs to involve additional assumptions on the sub-grid heterogeneity ("conceptualizations"). Likewise, meaningful physically based models also need to explicitly represent details of the landscape, and therefore need to be spatially distributed at larger scales, transferring knowledge inferred from observations across space.

It is worth noting that the term "physically-based" benefits from a misleading semantic-psychological bias. The term "physically based" wrongly implies that these models are *inherently* "correct" descriptions of real world-systems, which further implies the highly questionable notion that all other models are not "physical" and thus less "correct".

### 2.2.2 Conceptual models

Conceptual models provide a macroscale description of the hydrological system with a parsimonious and more abstract representations of the processes involved. Here the term macroscale is used to describe any scale larger than the scale of individual microscale observations used in physically based models. Zooming out to the macroscale therefore integrates natural microscale heterogeneities and feedback between them over the entire model domain, in spite of largely disregarding system-internal process complexity.

The basis of conceptual models are then relatively simple flux parameterizations to describe the large-scale manifestation of small-scale heterogeneities that emerge at the scale of the model application (e.g. catchment), as characterized by the

available integrated observations at that scale, such as stream flow. For that reason, conceptual models explicitly represent fewer individual hydrologic and, in particular, thermodynamic processes than physically based models (Figure 1). For example, evaporative processes are described by the empirical concept of potential evaporation in place of the detailed representations of the surface energy balance.

As many system boundary conditions and states cannot be directly observed at the macroscale, the flux parameterizations in conceptual models, e.g. in the form of Q=f(S), describe underdetermined systems and therefore require assumptions about their functional shapes and/or calibration of their parameters.

Conceptual models can be implemented as lumped or (semi-)distributed formulations (e.g. Kumar et al., 2010; Gao et al., 2014a; Fenicia et al., 2016). In spite of that they are sometimes collectively and inaccurately referred to as "lumped" models.

The terminology "conceptual model" itself to describe an abstract, macroscale representation of nature is really rather unfortunate as it is also used both by field scientists (e.g. McGlynn et al., 2004) and modellers (e.g. Gupta et al., 2012) to describe the understanding of the system. When viewed as abstract conceptual understanding, the "conceptual" model refers to all models, regardless of complexity, since all models are necessarily an abstract depiction of nature.

## 2.3 Model architecture: continuum based and bucket based

### 2.3.1 Continuum based models

Continuum based models equations are developed at the microscale and are applied directly on individual model elements. For the sub-surface a common continuum-based model is a 3-D implementation of the Richard's equation (e.g. Zehe and Blöschl, 2004; Kollet and Maxwell, 2006; Zehe et al., 2006; Sudicky et al., 2008). A distinguishing feature of continuum-based models is that model fluxes are computed based on spatial gradients in model state variables, e.g., flows are explicitly

computed based on the spatial gradient in matric head. Continuum-based models are hence inherently spatially distributed.

### 2.3.2 Bucket based models

Bucket or tank based models rely on "conceptual" storage elements ('buckets') to describe the storage and transmission of water through the flow domain. The buckets typically represent the storage of water at larger spatial scales, for example a hillslope or a catchment. The vertical and horizontal exchange of water between the buckets is then typically not expressed

by actual gradients, but rather, in a simplified way, exclusively as a function of the water storage in the conceptually-hierarchically "higher" bucket. For example the flux describing the infiltration from a bucket representing the unsaturated zone to a bucket representing the groundwater is often formulated exclusively as a function of the water storage in the unsaturated zone.

Lumped implementations of conceptual models are in general bucket-based. Yet, (semi-)distributed conceptual models can

involve simple, gradient-like controls on model internal exchange fluxes between buckets of individual model units (e.g.

hydrological response units), resembling simplified continuum-formulations (e.g. Weiler and McDonnell, 2004; Nijzink et al., 2016a).

## 2.4 Model refinement/scaling strategies: bottom-up and top-down

The distinction between bottom-up and top-down strategies describes rather broad modelling philosophies than specific approaches to formulate models.

### 2.4.1 Bottom-up models

The bottom-up scaling strategy often entails estimating large-scale fluxes by aggregating the output from individual, small-scale control volumes, i.e. the boundary fluxes (Beven, 2006a), along their respective surface and subsurface flow directions to the channel and eventually to the outlet of the system. As such, bottom-up approaches are rooted in inductive scientific reasoning: a set of (microscale) observations provides facts of the functioning of the system at that scale. Formulating theories that allow a meaningful integration of the small-scale observations (i.e. facts), pattern and general principles then emerge at larger scales (e.g. Andersen and Hepburn, 2016). In the absence of suitable observations, this aspect is commonly the bottleneck in hydrology, as many models rely on merely spatially aggregating fluxes to estimate fluxes at larger scales instead of actually integrating processes according to meaningful scaling relationships that account for the effect of heterogeneity, organization and feedback Thus, in spite of considerable success, the *inductive* approach to science in general and in hydrology in particular has in the past raised considerable criticism as it bases its conclusions on incomplete facts, therefore making them problematic to prove, i.e. the "black swan fallacy" (e.g. Popper, 1959).

Bottom-up approaches are typically accomplished using distributed, physically- and continuum based models (e.g. Kollet and Maxwell, 2008; Kumar et al., 2009; Camporese et al., 2010; Kollet et al., 2010; Maxwell et al., 2014; Piras et al., 2014), but strictly spoken, any kind of prediction or virtual experiment is necessarily a bottom-up approach.

### 2.4.2 Top-down models

The top-down approach to modelling is a hierarchal model refinement strategy that progressively tests and refines the model based on learning from data (Sivapalan et al., 2003). Crucially, the top-down approach is based on understanding and testing different models as competing alternative hypotheses of system functioning (e.g. Clark et al., 2011). With the aim to understand observed (macroscale) pattern by iteratively narrowing the range of possible system descriptions that can generate these observations and which are typically assemblages of various individual mechanisms, the top-down approach is therefore a reflection of *deductive* scientific method (e.g. Popper, 1959; Salmon, 1967).

Based on observations of system integrated variables, such as stream flow, top-down modelling applications attempt to describe the system directly at the scale of the system, which in hydrology frequently is the catchment-scale (Klemes, 1983; Dooge, 1986). However, the approach can in principle be applied at any desired scale. For example, to understand which individual mechanisms, including for example the effects of pore size distributions, particle charge density or viscosity

distributions, are necessary to describe what emerges as Darcy's law at the soil sample scale. The top-down approach is criticized for lacking generally valid criteria for rejection of hypotheses and for its dependence on rigorous testing procedures, which are unavailable in reality due to the absence of sufficiently detailed observations.

Being an iterative process, top-down approaches typically start with simple spatially lumped, conceptual, bucket based models, but can, in principle, subsequently involve model formulations at any point along the resolution-complexity continuum. Some examples of studies applying the top-down approach include Young (1998, 2003), Jothityangkoon et al. (2001), Kon and Sivapalan (2007), Fenicia et al. (2008, 2016), Kavetski and Fenicia (2011), Gharari et al. (2014a), Hrachowitz et al. (2014), Willems (2014) or more recently Garavaglia et al. (2017).

## 3 Modelling myths – or not?

There is a wide range of frequently communicated beliefs and assumptions on alternative approaches to modelling. They reflect different perceptions of modelling limitations. In the following sections we will contrast and scrutinize modelling critiques commonly communicated by the two respective modelling communities, discuss the extent to which we believe they are justified, describe how different strengths of different approaches are complementary and how combining them may benefit model convergence and eventually improved predictions.

### 3.1 Critique: Physical basis

#### 3.1.1 *"Bucket models have a poor physical and theoretical basis."*

Since bucket models originate from empirical approaches to mimic the hydrological response based on observations at the macroscale, such as stream flow, without further assumptions on the system internal processes, this statement does certainly have an element of truth. However, evaluating this statement requires considering the effects of scale, organization and emergent properties of a system.

Models based on macroscale observations seek to describe the system integrated observed response without loss of essential information. There is no loss of information, in theory, because the effects of sub-element information (e.g. heterogeneity) remains implicitly encapsulated in the large-scale functional relationships between model states and model fluxes. In general, water flows follow the observable, physical phenomenon of spatio-temporal dispersion of discrete input signals, controlled by water and energy input, gravity, flow trajectories and flow resistances (e.g. Rinaldo et al., 1991; Snell and Sivapalan, 1994; Robinson et al., 1995; Botter and Rinaldo, 2003). A hydrological system, e.g. a catchment, therefore constitutes a low-pass filter. It disperses a random input signal (i.e. precipitation) by buffering its high-frequency components in storage components and by eventually releasing it with a suite of system specific time lags as stream flow or evaporation. Being in the realm of organized complexity (Dooge, 1986), the hydrological response at the catchment-scale can in most cases not be fully described by exclusively statistical methods and thus by the simplest bucket models, such as single linear reservoirs, or related concepts such as the Instantaneous Unit Hydrograph (Sherman, 1932). Typically adopting

a top-down approach, the development of bucket models is then the process of meaningfully representing the large-scale manifestation of organized complexity, introduced by spatial heterogeneity, by identifying a range of different dominant functional relationships between system input and the integrated output emerging through organization at the macroscale, i.e. the testing of competing hypotheses (e.g. Clark et al., 2011; Fenicia et al., 2011), without the need of resorting to small scale

physics.

In spite of being mostly conceptual in their design and the associated high level of abstraction, bucket models satisfy conservation of mass and typically provide a conceptual, parsimonious representation of the energy balance based on the concept of potential evaporation. The energy balance can be approximately closed if the model is carefully constrained not only with respect to the hydrograph but also with respect to the actual evaporation. In the common absence of more detailed

observations, such energy balance constraints can be imposed using observed runoff coefficients on a range of scales (e.g. annual, seasonal and event-based), which define the partitioning between streamflow and evaporative fluxes (e.g. Budyko, 1974; Donohue et al., 2007; Sivapalan et al., 2011) plus potential deep infiltration losses (e.g. Andreassian and Perrin, 2012). Notwithstanding its value, this strategy also illustrates one of the main weaknesses of many conceptual, bucket models: the lack of a more detailed representation of the energy balance only allows to approximate longer-term conservation of energy

but does not continuously guarantee it over shorter time scales. In addition, the concept of potential evaporation effectively partitions net radiation into sensible and latent heat fluxes but does not explicitly track the residual energy that is not used for evapotranspiration such as the feedback between the potential and the capillary binding energy of water or the export of kinetic energy in water fluxes leaving the system.

The purported physical basis of macroscale laws permits that a physical meaning can (and actually should eventually) be

assigned to all processes in (conceptual) bucket models. Purely data-driven developments of bucket models, resembling signal processing approaches and thus understanding the hydrological system merely as a mathematical low-pass filter whose properties need to be identified, mostly forgo this process (e.g. Young, 2003). In contrast, for models that were developed with a mind-set directed more towards actual process understanding, the hydrological function of individual model components has in the past often been casually and loosely "interpreted". However, without detailed testing, such

interpretations of their physical basis remain somewhat ambiguous and subjective.

To strengthen the physical basis, it will eventually be necessary to explore methods to more objectively and rigorously test individual model sub-components against observations (Clark et al., 2011) and/or to assign physical meaning to them *a priori* (cf. Bahremand, 2016). A potentially effective starting point for the latter is to use observations at the modelling scale to infer information about the functional shapes and to quantify the actual parameters of individual processes at that scale.

Examples include the concept of Master Recession Curves (Lamb and Beven, 1997) or the water holding capacity in the unsaturated root zone ($S_{U,max}$), which is the core of many hydrological systems as it controls the partitioning of drainage and evaporative fluxes (Gao et al., 2014b; deBoer-Euser et al., 2016; Nijzink et al., 2016b). These system components integrate heterogeneities and quantify actual physical properties present and physical processes active at the observation and modelling scale. Providing clear physical meaning to different parts of a model will then necessarily constrain the feasible

model space and consequently increase a model's hydrological consistency while reducing its predictive uncertainty (cf. Kirchner, 2006).

We therefore argue that bucket models developed based on deductive scientific reasoning, do have, if well implemented and tested, a robust physical and theoretical basis at the macroscale, and that it is possible to relate their individual
components to stores and fluxes in nature (e.g. Clark et al., 2008; Fenicia et al., 2016; Gao et al., 2016), albeit at a different spatial scale and process resolution than continuum-based models. These types of models emphasize the value of zooming out and understanding the system from the point of holistic empiricism. Potential ways forward to better exploit the potential of these models may involve explicit treatment of the energy balance as well as detailed observation-based process identification.

**3.1.2 *"Continuum-based models are applied at scales for which their equations were not developed"***

Continuum-based models are typically distributed, physically based models, frequently developed with a bottom-up approach. The general theory behind the fundamental equations of such models is based on forces acting on and fluxes passing through infinitesimal control volumes. This implies homogeneity over the entire control volume and allows the assumption of a local equilibrium (i.e. well mixed conditions), which is necessary for a meaningful definition of potential
gradients. However, it was shown that the assumption of local equilibrium does not hold at scales above one meter (e.g. Or et al., 2015), which is exacerbated by the absence of suitable observations to formulate up-scaling relationships that allow a meaningful representation of emergent processes at larger scales. The Darcy-Richards formulation further poses that water movement in porous media is (1) controlled by equal flow resistances for both, gravity and capillarity driven fluxes and (2) exclusively characterized by diffusive fluxes and thus by the absence of kinetic energy. These assumptions may not be
suitable to describe fluxes during wet conditions, which in many systems are characterized by an increased importance of advective, and thus velocity- rather than celerity-driven processes (e.g. McDonnell and Beven, 2014). As a consequence, the small-scale equations do not necessarily represent the large-scale impact of sub-grid-scale heterogeneities (Beven, 1989), and the spatial gradients in model state variables do not have much meaning at the spatial resolution of the model (e.g. 1-km grids).

On the other hand, continuum-based models are also criticized because there is insufficient data to reliably describe the spatial heterogeneity of the storage and transmission properties of the sub-surface. Being a non-linear system, for example averaging observed point-scale van Genuchten parameters, does not result in a meaningful representation of the average water retention characterization for larger-scale model elements.

These two issues are linked and exacerbated by the problem that the higher the spatial resolution of the model domain, the
higher the number of exchange fluxes (i.e. boundary fluxes; Beven, 2006) between individual adjacent modelling units in the model. Increasing the degrees of freedom in a model, this leads to the situation in which a specific choice of model parameters, no matter if observed or calibrated, remains problematic to test against observations.

A potentially valuable way forward to somewhat circumvent the above points may be to relax the assumptions required by the Darcy-Richards equation and to replace the rigorous formulation with some degree of scale-independent conceptualization (e.g. Craig et al., 2010). For example, instead of averaging van Genuchten parameters, the ensemble of actual observed water retention curves at different locations in the system could be used to estimate upper and lower bounds of effective pedotransfer functions. As recently illustrated by Loritz et al. (2017), integrating some of the heterogeneity in such a way, these functions may be more representative for larger areas. This in turn allows to reduce the spatial resolution of the model domain and the associated problems.

## 3.2 Critique: Natural heterogeneity and model complexity

### 3.2.1 *"Conceptual models are too simplistic and cannot adequately represent natural heterogeneity."*

Simple lumped conceptual models, such as HBV, have a long track record of, at first glance, successful applications in a wide range of catchments world-wide. However, this success is in many cases deceptive as these models are often used in a quasi-inductive way with an implicit *a priori* assumption that they are a meaningful representation of the system, thereby not treating the model as a hypothesis and not testing alternative formulations.

The importance of adequate representations natural heterogeneity is largely undisputed (e.g. Clark et al., 2011; Gupta et al., 2012). However, frequently model calibration is (unnecessarily) limited to time series of streamflow observations which provides merely insight into a very small number of parameters (Jakeman and Hornberger, 1993). Thus, although any additional model process has the potential to improve the representation of heterogeneity, the required additional calibration parameters increase the feasible model (or parameter) space and the resulting potential for equifinality (Beven, 1993), thereby turning models into the oft-cited "mathematical marionettes" (Kirchner, 2006). In spite of its skill to reproduce the calibration objective, such a model will in many situations struggle to simultaneously reproduce different additional system internal dynamics (e.g. groundwater fluctuations) and emerging patterns (e.g. flow duration curves), indicating its failure to meaningfully represent dominant processes and their heterogeneity in a catchment, which in turn often results in a poor predictive power of these models. This was in the past demonstrated by many studies (e.g. Jothityangkoon et al., 2001; Atkinson et al., 2002; Fenicia et al., 2008; Euser et al., 2013; Coxon et al., 2014; Fenicia et al., 2014; Hrachowitz et al., 2014; Willems, 2014).

The lack of an adequate model calibration, testing and evaluation culture partly arises both from insufficient exploitation of the information content of the available data, but also from the real lack of suitable data (Gupta et al., 2008; Clark et al., 2011). Under these conditions, many models remain ill-posed inverse problems. To limit the associated equifinality, Occam's razor is commonly invoked to make models "as simple as possible but not simpler" (e.g. Clark et al., 2011). But how simple is "as simple as possible"? In other words, how large a model space (i.e. possible parameterizations and prior parameter space) can be constrained with available information to identify reasonably narrow posterior distributions while

ensuring a high as possible multi-objective and multi-variate model performance? To analyse this, the two axes of the spatial resolution-process complexity continuum (Figure1) need to be considered separately.

The required *spatial resolution* for a model to represent the major effects of heterogeneity on the hydrological response does not only depend on the degree of surface and sub-surface heterogeneity, but also on the hydro-meteorological

conditions in the region of interest, as shown in an illustrative example in Supplementary Material S1. Briefly, in cool, humid and thus energy-limited regions the level of water storage can remain elevated throughout the year, thus providing only limited storage capacities. In such a situation, many of the processes that introduce non-linearity, e.g. through spatially heterogeneous thresholds, and thereby control the emergence of hydrologic connectivity are not dominant or even negligible. This is in contrast to warm, arid and thus water-limited regions, where heterogeneous storage deficits over the model domain

will exert much stronger and often spatially heterogeneous controls on the hydrological response. In summary, lumped conceptual can be suitable macroscale representations of hydrological systems in some regions, while in other regions more spatial discretization is required. The relevant questions are: How do different heterogeneities affect water storage and release in different environments? Which types of heterogeneity can be captured by a single emergent functional relationship and for which types several functional relationships at the macroscale are necessary to meaningfully describe observations?

*Process complexity*, i.e. the detail to which models explicitly represent specific processes in terrestrial hydrological systems is, at its fundamental level, characterized by two major partitioning points that control how water is stored in and released from systems through upward, downward or lateral fluxes (e.g. Rockström et al. 2009; Clark et al., 2015; Savenije and Hrachowitz, 2017). Near the land surface, precipitation is split into (a) evaporation and sublimation from vegetation and ground surface interception (including snow) as well as from open water bodies, (b) overland flow and (c) infiltration into

the root zone. Water entering into the root zone, is further partitioned into (d) soil evaporation, (e) plant transpiration, (f) shallow, lateral subsurface flow through features such as shallow high permeability soil layers, soil pipe networks or a combination thereof and (g) percolation to the unsaturated zone and the groundwater *below* the root zone.

As emphasized by Linsley (1982), all fluxes (a-g) are present in essentially any catchment, albeit with different relative importance in different environments, and therefore need to be represented in a model. This can be illustrated with the

occurrence of weather events that are uncommon for a specific region. In the Atacama Desert, one of the driest places on earth with little or no vegetation under average conditions, uncommonly high spring precipitation, such as in 2015, can cause episodic appearance of abundant vegetation. This temporally changes the partitioning pattern and thus the hydrological functioning of the region as plant transpiration that is otherwise absent is "activated". Similarly, rare occurrences of snow fall can cause temporal anomalies in the hydrological functioning of otherwise warm regions, such as 2013 in the Middle

East. In spite of them being "de-activated" most of the time, such processes are in principle present and need therefore also be conceptually reflected in any hydrological model structure. However, if considered negligible in a specific environment during a modelling period of interest, the modeller can decide to deactivate individual processes by using informed prior parameter distributions. In other words, the respective parameters will be set to suitable fixed values that effectively switch off the process using Dirac delta functions as prior distributions.

The key decision for the modeller is then to decide to which level of detail the individual processes at the two partitioning points will be resolved and how they can be parametrized (cf. Gupta et al., 2012). The questions to be answered are: How much detail is *necessary* to reproduce observed dynamics and pattern? How much detail is *warranted* by the available data to meaningfully parameterize and test the chosen process representation? An example to illustrate the thought process involved is provided in the Supplementary Material (S2).

Conceptualizing the hydrological system by zooming out and explicitly representing only dominant processes by exploiting simple functional relationships (or pattern) emerging as a result of organization at the macroscale (e.g. Ehret et al., 2014) has the advantage of significantly reducing the number of required effective model parameters. Importantly, this lumping process does not, as long as it is well tested to encapsulate the relevant dynamics of the system, necessarily involve a loss of information. It should therefore not be understood as "simplification" of the system. Rather, it has the potential to *integrate* the interaction of heterogeneous processes at the microscale over the entire domain of interest and thereby to provide a system description that is consistent with real world observations at the scale of interest without the need for further assumptions and the related uncertainties.

It is true that untested and poorly evaluated applications of standard lumped conceptual models are often oversimplifications that do not adequately reflect natural heterogeneity and its effects on the hydrological response. However, conceptual models can be formulated at any level of process and spatial complexity, limited only by the available information. The actual problem is therefore not the conceptual model *per se* but rather the way it is implemented and applied. The decision, which degree of zooming out, i.e. which level of detailed process representation is feasible and which level is necessary, eventually needs to be made by the modeller on basis of the available observations, acknowledging that *all* hydrological models at the catchment scale are to a certain extent conceptualizations. When carefully implemented, spatially distributed formulations, e.g. based on hydrological response units or related concepts (Beven and Kirkby, 1979; Knudsen et al., 1986; Flügel, 1995; Winter, 2001; Seibert et al., 2003; Uhlenbrook et al., 2004, 2010; Schmocker-Fackel et al., 2007; Gharari et al., 2011; Zehe et al., 2014; Haghnegahdar et al., 2015), with an equilibrated balance between process heterogeneity and information/data availability and tested and evaluated against multivariate observed response dynamics, conceptual models have been shown to be versatile enough to identify and represent the dominant hydrological processes and their heterogeneity in a catchment (e.g. Boyle et al., 2001; Fenicia et al., 2008a,b; Winsemius et al., 2008; Samaniego et al., 2010; Kumar et al., 2013; Hrachowitz et al., 2014; Nijzink et al., 2016a) within limited uncertainty.

### 3.2.2 *"Physically based models are too complex and give a deceptive sense of accuracy"*

Mirroring the statement that conceptual models are too simplistic and do not represent heterogeneity, it may in a similar way be valuable to discuss the question if, in the absence of appropriate observations at the scale and resolution of interest, distributed, physically based models with high process and spatial complexity are not too complex and somewhat deceptive about the accuracy that is implied by their formulation.

Physically based models are frequently developed and applied under the implicit up-scaling and bottom-up premise that the heterogeneous system boundary conditions and thus the model parameters are known from observations and representative for the scale of the modelling units of a given model. However, three, partly related points make this assumption problematic for many model applications: (1) the spatial resolution of observations, (2) the spatial scale of observations and (3) the accuracy of the observations.

There is often insufficient geophysical information to represent the heterogeneity of the subsurface over large domains relevant for water resources planning and management. For example, the low spatial resolution of many available soil maps may incorrectly indicate that the storage and transmission properties of soil are spatially homogenous, i.e. a single soil type over an individual modelling unit or even over an entire catchment. Similarly, data on the root-systems of vegetation, used to estimate an important source of system non-linearity and thus one of the core parameters in a model, the storage capacity in the unsaturated root zone, is, at best, available for a few individual plants. As such it does not sufficiently account for distinct effects caused by different ecosystem compositions in different parts of the system (e.g. different mixtures of species), age distribution of the plants in the system, the density of plants (i.e. individual plants per unit area) or, being mostly snapshots in time, temporally evolving root-systems (deBoer-Euser et al., 2016; Nijzink et al., 2016b; Savenije and Hrachowitz, 2017). In addition, the available meteorological forcing data may be overly smooth and/or unrepresentative due the methods used to interpolate station data from sparse observing networks.

Related to the spatial resolution is the spatial scale of the available observations. Many model parameters are directly inferred from observations at small-scales, e.g. core sample or plot-scale, assuming they are representative for the, often much larger, respective modelling unit. This is critical e.g. for the determination of soil hydraulic conductivities or the water retention curve, as the small scale of the observations may often fail to meaningfully characterize larger features in the soil matrix, such as macropores, together with their spatial distribution.

Finally, with increasing complexity, non-linear systems become increasingly problematic to predict with detailed, small-scale descriptions, due to uncertainties in the necessary observations of boundary conditions, forcing and system states (e.g. Zehe et al., 2007) caused by the combined effects of limited observation accuracy and representativeness.

Spanning several orders of magnitude in scale, from the microscale (e.g. soil particles) to the continental scale (e.g. mountain ranges), it is unlikely that observation technology will ever enable a comprehensive and non-invasive description of the heterogeneity in hydrologic systems, especially for large model domains (Refsgaard et al., 2010).

From that perspective, it is not unreasonable to argue that many implementations of distributed, physically based models are somewhat over-ambitious and overly optimistic given our actual knowledge of the system as their degree of spatial resolution and/or process complexity is, strictly spoken, not warranted by the available data. This is in particular true for applications that make direct use of scarce small-scale observations and, in spite of the associated limitations, fail to provide a meaningful uncertainty analysis. As shown in the illustrative example in the Supplementary Material S2, each process represented in a model, no matter at which scale, is a larger scale manifestation of the integration of the interactions of individual heterogeneous processes at yet smaller scales, down to molecular levels (or perhaps even beyond). This implies

that there is no "natural" cut-off point at which all processes in the system are completely represented. All process descriptions in a model thus involve at least some degree of conceptualization, making use of functional relationships emerging at larger scales.

The relevant question therefore is, up to which level we can zoom out and integrate individual processes into conceptual functional relationships at larger scales, without losing information and thereby benefitting from a reduced dimensionality of the parameter space. This question is tightly linked to the question which spatial resolution and process complexity is required to answer questions relevant for water management purposes in specific cases. In other words, apart from being theoretically satisfying, do we actually *need* to discretize a catchment e.g. into 1-cm grids for a model to be a useful tool?

The above points do however not contest the immense value of physically based models as recently discussed in detail by Fatichi et al. (2016). Rather, detailed implementations of these models, in spite of the associated limitations, have in the past been shown to be powerful tools to reproduce and understand spatially heterogeneous system-internal flux and state dynamics as well as patterns that emerge from the interaction of small-scale processes (e.g. Kollet and Maxwell, 2008; Maxwell and Kollet, 2008; Vivoni et al., 2010; Bearup et al., 2014; Sutanudjaja et al., 2014). As such they are very well suited for virtual experiments at a range of scales (e.g. Ivanov et al., 2010; Fatichi et al., 2014; Bierkens et al., 2015; Maxwell et al., 2015;2016). This is in particular true to understand and assess the impact of disturbances such as climate and/or land use change in scenario analyses, as systemic change or even tipping points can emerge from changes in one or more individual small-scale model components and the associated feedback (e.g. Maxwell and Kollet, 2008; Bearup et al., 2016) .

## 3.3 Critique: Hypothesis testing and calibration

### 3.3.1 *"The top-down modelling approach successively evaluates ad-hoc formulations of untestable hypotheses"*

It is important to realize that the top-down approach is a modelling strategy and not a specific model formulation. In spite of that, many applications of conceptual bucket models, are falsely referred to as "top-down models", while being mere and unquestioned applications of off-the-shelf models, such as HBV or FLEX. Such insufficient model testing and *ad-hoc* model applications implicitly assume that these models can adequately represent observed hydrological response dynamics in different catchments, thereby violating the fundamental requirement of top-down approaches: the testing of alternative hypotheses. It largely ignores that any model is an assemblage of hypotheses consisting of individual building blocks and their parametrizations, encapsulating the modeler's understanding how a specific environment shapes the hydrological system. The point is that different environmental conditions dictate the need to test if the prior information on the parameters needs to be changed and/or relaxed so as to activate a process that was deactivated in a model previously used in other environments (or vice-versa) to adjust the model to the prevailing environmental conditions.

A meaningful decision on the use of given prior parameter distributions and their information content for a model application in a specific environment can only be made if the model hypothesis is carefully tested. However, it is sometimes

argued that entire models are untestable hypotheses, as they represent an ensemble of different processes or parts of the system. Models, therefore, need to be seen as sets of distinct hypotheses that need to be tested independently to avoid the adverse effects of equifinality (Clark et al., 2011). Recalling the above argument (see section 3.2.2) that when disaggregating a system, the pattern emerging at each subsequent level of detail result from interacting heterogeneous processes at yet smaller scales. Thus, down to that level, every hypothesis consists of several other, smaller scale hypotheses. The relevant question arising here is, to which level do model components then have to be disaggregated to constitute testable hypotheses? Thus, of course, treating a model as a single hypothesis does not make the hypothesis *untestable*. Rather, given the system-integrated nature of many observations and the frequently limited number of performance indicators considered to test the model against, it may in many cases remain a relatively *weak* test. In contrast, individually testing sub-components of the system will provide the modeler with more information because its sub-components are necessarily less complex than the overall model. This, in turn, provides less possibilities for compensating misrepresentations of one process by wrongly adjusting other processes. In other words, it will have higher potential to avoid Type I errors (i.e. false positives), therefore resulting in a stricter test. The obvious problem arising here is less of theoretical than of practical nature: besides epistemic uncertainties, observations of system sub-components, including the often cited boundary fluxes (Beven, 2006a), to test the model components against are typically not available at the scale and/or resolution of interest or not available at all, although with the ever improving spatio-temporal resolution and quality of remote sensing products the problem will potentially be somewhat alleviated in the near future. Clearly, from that perspective, weak model tests are in the frequent absence of other options preferable to no tests at all. Given these practical constraints for model falsification, systematic and exhaustive multi-objective and multivariate calibration strategies and post-calibration evaluation procedures need to be part of any top-down modelling approach to ensure that the overall modelled system response, including emerging patterns (e.g. flow duration curves), reproduces the observed response dynamics in a meaningful way. The above point is very closely related to the necessity of calibration. If the system could be observed as a fully controlled system at the scale and resolution of interest (e.g. catchment scale for lumped models, grid scale for distributed models), there would be little additional need for testing as the system would be well constrained and its functioning well understood. Thus, much of the problems discussed above is a direct consequence of the absence of such observations. Whenever no adequate observations are available, a model that aims to represent a specific real world system requires calibration. Any model.

We therefore argue that top-down modelling approaches do not evaluate "ad-hoc formulations of *untestable* hypotheses" but rather that many hypotheses often remain *untested*. The actual problem therefore not being the model strategy ("top-down") or type ("conceptual bucket"), but the way these models are frequently applied in a careless way. This is exacerbated by the fact that in the past only a few studies attempted to develop a general framework for objective and science-based model selection (e.g. Young, 2003) and thus a general and systematic approach to learning from data (Sivapalan et al., 2003).

### 3.3.2 *"Physically based models have too many degrees of freedom and cannot be meaningfully constrained"*

As argued above, detailed distributed implementations of physically based models to represent specific real world systems may provide a deceptive sense of accuracy if applied as a bottom-up approach and thus operated with highly informed prior parameters distributions (e.g. fixed parameter values or regularized estimates), based on anecdotal, point or plot scale observations that do not match the scale and resolution of the individual modelling units (e.g. grid cell). In such a case, to avoid misrepresentations of the system, parameter values effective at the scale of the model grid cells need to be selected otherwise, typically by calibration. The high degree of freedom in the model, however, will result in considerable equifinality.

Even if there was an adequate correspondence of the respective scales of field observations and modelling units two further problems remain: observations of both, system boundary conditions as well as system states (e.g. groundwater levels) or fluxes (e.g. evaporation) are typically, if at all, available at low spatial resolution. This implies (1) that the boundary conditions in the remainder of the system are unknown and that its heterogeneity is very likely to be misrepresented in a model and (2) that modelled system states (e.g. groundwater levels) or fluxes (e.g. evaporation) can only be tested against observations for a small number of modelling units, thereby only providing a weak test for the model.

Although the above limitations are in principle valid, it has previously been shown that uncalibrated, physically based models, operated with parameters from direct observations, can meaningfully and simultaneously reproduce different aspects of the hydrological response (e.g. Maxwell et al., 2015). Fatichi et al. (2016) argue that this suggests that uncertainties in observed system input and output data and the resulting biased parameters in calibrated models (e.g. Renard et al., 2010) outweigh uncertainties introduced by insufficient heterogeneity and/or unsuitable scale.

The inherent strength of physically-based models (see section 3.2.2) is their spatially explicit and detailed formulation of processes which allows the analysis of emergent pattern to the system, in particular after disturbance scenarios within virtual experiments, leading to a better understanding of the system overall behaviour. However, we think that, as every model is a simplification of reality (Gupta et al., 2012), even physically based models should, if used for actual hydrological predictions in specific systems, be treated as hypotheses and thus subject to testing and evaluation procedures. For example, relaxing, to some degree, the information on the prior parameter distributions, i.e. replacing fixed parameter values with reasonably narrow prior distributions, will allow more flexibility and may therefore, in a testing procedure allow the identification of parameters that provide a more suitable representation of the system (Mendoza et al., 2015). Given the high dimensionality of the parameter space, this clearly also entails the need for additional model constraints beyond traditional calibration. Apart from the use of similar multiple objective functions and multiple flux and state variables for the evaluating the model against observations as used for conceptual models, the use of regularization (e.g. Pokhrel et al., 2008; Samaniego et al., 2010), data assimilation (e.g. Shi et al., 2014) and similar techniques (e.g. Refsgaard et al., 2006) has proven helpful to identify feasible model parameters. Detailed physically based models, furthermore, offer the opportunity to fully exploit the value of additional and *simultaneous* evaluation against remotely sensed spatio-temporal pattern, such as snow pack dynamics using

MODIS snow cover data (e.g. Kuchment et al., 2010), estimates of water storage anomalies using GRACE (e.g. Syed et al., 2008) and many others.

## 4 Implications

From the above discussion, a few relatively clear and unambiguous points define the basis, functioning and limitations of competing approaches for process-based hydrologic modeling. Condensing these points, it emerges that:

(1) All hydrological models are to some extent "conceptual" and to some extent "physical", they largely only differ in the degree of detail they resolve the system, which in turn is dictated by the available data. Conceptual bucket models approach the problem from a macroscale physical understanding, while physically based continuum models emphasize the microscale perspective. An ideal model would, almost needless to say, provide good representations of both aspects.

(2) Modelling strategies starting at opposite ends and follow a gradual transition along the resolution-complexity continuum (Figure 1). While conceptual bucket based models constitute a physically based approach to hydrological modeling that is rooted in holistic empiricism, similar to statistical physics, physically based continuum models are based on mechanistic descriptions of small-scale physics.

(3) Different modelling strategies are complementary rather than mutually exclusive as they have different strengths and are thus suitable for different purposes. While conceptual bucket based models have advantages for operational predictions of specific real world systems, physically based continuum models may in many cases be preferable for more generally explaining multi-causal relations in terrestrial systems, in particular the spatio-temporal impacts of disturbances.

(4) All models can, in principle, be implemented with any desired detail. The key question is whether additional process complexity can be tested against and is justified by the available data. This is true for both process and spatial complexity, which also highlights that we are really crossing a continuum of complexity, where conceptual bucket models converge towards physically based continuum formulations.

(5) All models must reflect our conceptual understanding of the system in regards to how water fluxes are partitioned at the near-surface and the unsaturated root zone. Since all relevant fluxes can be present in any environment, albeit with different relative importance, all models therefore need to have the same fundamental model structure (but not necessarily the same parameterization) to reflect these processes.

(6) In the absence of sufficient observations at the modelling scale and resolution, all hydrological models remain hypotheses and require rigorous testing and post-calibration evaluation if used to represent specific real world systems.

(7) All hydrological models applied at scales beyond the plot scale and if used to represent specific real world systems require some degree of calibration, as direct observations of effective parameters at these modelling scales and

resolutions are typically not available. Improper application of parameters from observations that do not match the modelling scale and/or resolution may not provide a sufficient representation of the natural heterogeneity of this parameter can lead to misrepresentations of the system and give a deceptive impression of accuracy.

(8) The fundamental problems in catchment modelling do not lie in the type of model used, but rather in the way a model is applied.

## 5 Steps towards convergence of modelling strategies

Taken together, the above arguments suggest that the perceived and somewhat arbitrary dichotomy between different modelling strategies leads to some degree of confusion. Acknowledging that all models are to some degree conceptual, and that often not the actual models are the problem but the inadequate way they are applied, may open up the view towards the real fundamental questions in catchment-scale modelling: how much detail do we *need* in our models and how much detail is *warranted* by data for different applications? To find a balance that allows us to best describe the system based on scientifically robust grounds and thus a way towards a convergence of different modelling strategies will benefit from exploiting the features of macroscale organization and pattern formation as well as from adopting a general culture of rigorous hypotheses testing.

## 5.1 Organized complexity and catchment similarity

Progress in catchment-scale understanding of hydrological functioning and the related development of models for more reliable predictions hinge on a better understanding of how natural heterogeneities at all scales aggregate to larger scales and how this influences the hydrological response. As already emphasized previously by many authors (e.g. Beven, 1989,2001,2006a; Kirchner, 2006; Zehe et al., 2014), these efforts to approach the closure problem in hydrology need to involve both, ways to reliably determine effective model parameters, i.e. the system boundary conditions, that integrate and reflect the natural heterogeneity within the model domain, as well as the development of equations that are physically consistent at the scale of application. These scale and heterogeneity issues were acknowledged already in the early 1980s to be at the core of many problems for our understanding and modelling of hydrological systems (e.g. Dooge, 1986; Wood et al., 1988; Wood et al., 1990; Blöschl and Sivapalan, 1995; Blöschl, 2001). It was, for two decades or so, indeed a very active and fruitful field of research but it has somewhat lost momentum. Ten years after the landmark papers of Beven (2006a) and Kirchner (2006), remarkably little progress was made and many ideas and concepts did not find their way into mainstream hydrology. Nevertheless, it is imperative to understand how processes scale, heterogeneity aggregates and how this controls the emergence of patterns at the large scale. This then has the potential to enhance our understanding of what controls catchment functioning and our ability to develop models (e.g. Vinogradov et al., 2011). A potential way forward towards achieving this, may be the much advocated large sample, comparative hydrology and similarity analysis to identify pattern and generally applicable, functional relationships (e.g. Sivapalan et al., 2003; McDonnell et al., 2007; Blöschl et al., 2013;

Sivakumar et al., 2013; Gupta et al., 2014). Recently receiving increased attention (e.g. Lyon and Troch, 2007; Carillo et al., 2011; Sawicz et al., 2011; Coopersmith et al., 2012; Berghuijs et al., 2014,2016; Fenicia et al., 2014; Li et al., 2014; McMillan et al., 2014), using similarity analysis to improve our understanding of the link between catchment structure and hydrological functioning at the macroscale will be instrumental to guide the development of meaningful model hypotheses and to constrain the feasible parameter space in a way that forces the model to reproduce these characteristics emerging at the macroscale. A recent example includes Ye et al. (2012) who identified dominant process controls underlying regional differences in regime- and flow duration curves. Similarly, Gao et al. (2014b) demonstrated how the model parameter representing the water storage capacity in the unsaturated root zone at the macroscale can be considerably constrained exclusively based on water balance data.

## 5.2 Spatial patterns

Being one of the  main advantages of most physically based continuum models, the value of representing spatial differences in model fluxes and states as manifestations of heterogeneities in the system, is considerably under-exploited in conceptual bucket models. It is well established that hydrological connectivity exhibits not only temporal but also spatial dynamics and that therefore source areas of flow generation vary over time (e.g. Lehmann et al., 2007; Spence et al., 2010; Jencso and McGlynn, 2011; Ogden et al., 2013). Adapting the spatial resolution of models to the spatial resolution of available observations offers considerable potential to improve the representation of process dynamics across the model domain. This is in particular true as observations of spatial pattern in one or more variables, such as snow cover or evaporation, can then be used as additional model constraints to offset the adverse effects of increased degrees of freedom (e.g. Immerzeel and Droogers, 2008; Xu et al., 2014; Lopez et al., 2017).

## 5.3 Models as hypotheses

There is a clear need to establish a mainstream culture of robust model calibration and rigorous post-calibration testing/evaluation of alternative model formulations (i.e. hypotheses) for any type of model. Such work is necessary to achieve progress in catchment-scale modelling and advance the use of models as scientific tools.

Stronger and more meaningful model tests with respect to multiple variables, model states and hydrological signatures need to become a standard procedure (e.g. Willems et al., 2014; Clark et al., 2015) as it was previously shown that, although models frequently exhibit considerable skill to reproduce the hydrograph during both, calibration and "validation", many of these models struggle to reproduce other system relevant features. This includes, for example groundwater table fluctuations (e.g. Fenicia et al., 2008), long-term average runoff coefficients as a proxy of average actual evaporation (e.g. Gharari et al., 2014b; Hrachowitz et al., 2014) and solute dynamics (e.g. Birkel et al., 2010; Fenicia et al., 2010) as well as hydrological signatures of the system, e.g. duration curves (e.g. Euser et al., 2013; Kelleher et al., 2017). In addition, model calibration and/or evaluation against observed spatial pattern remains currently still under-exploited.

In spite of the computational costs involved, we argue that development of detailed physically based continuum models would also strongly benefit from adopting more of a top-down perspective. This would be beneficial for, in particular, highly conceptualized model components, such as, but not limited to, those related to preferential flow.

In any case, comprehensive model calibration and/or testing strategies have the potential to identify and reject models (i.e. parameters and parameterizations) that "do not meet minimum requirements" (Vache and McDonnell, 2006), which in can considerably reduce type I errors, i.e. falsely accepting poor models when they should be rejected ("false positive"; Beven, 2010).

## 5.4 Model uncertainty

We are currently in a position where we, in an exaggerated way, feed wrong models with wrong input data and calibrate them to wrong output data to obtain wrong parameters. In the light of so many unknowns, comprehensive, systematic, end-to-end uncertainty analysis needs finally to become a standard component of any modelling study (e.g. Beven, 2006b; Pappenberger and Beven, 2006). This applies to any type of model. However, as full uncertainty analysis of physically based continuum models may remain computationally challenging for the foreseeable future, it may be worth to consider reporting results in model ensembles, similar to what is common practice, for example, in atmospheric sciences. In any case, systematic uncertainty analysis has the potential to significantly reduce type II errors, i.e. rejecting a good model when it should have been accepted ("false negative"; Beven 2010) and is thus instrumental to avoid giving a false impression of accuracy in our models.

## 6 Concluding remarks

On balance, we believe that modelling of catchments will significantly benefit from and may even require a convergence of different modelling strategies, in particular with respect to exploiting the features of organization in these complex systems (Dooge, 1986) in a hierarchical way, as for example suggested by Zehe et al. (2014). Combining this with large sample comparative studies and more efficiently exploiting the information content of available data over a range of scales may eventually pave the way towards the formulation of meaningful and general scaling relationships that will be key to understand how large scale pattern emerge from first principles. The aim has then to be the development of models that are linked by these scaling relationships across the entire resolution-complexity continuum so that zooming out from detailed, microscale representations, observed large scale patterns are reproduced and, vice versa, that discretizing macroscale representations will result in meaningful representations at the microscale. In that sense we would like to strongly encourage researchers to not only acknowledge but to actively make use of a diverse range of modeling strategies in order to strengthen their own models.

**Acknowledgements**

We thank the editor Erwin Zehe as well as Marc Bierkens, Hoshin Gupta, Ralf Loritz, Sivarajah Mylevaganam and Thorsten Wagener for their critical yet very constructive comments and suggestions that helped to considerably improve this manuscript.

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

**Supplementary Material**

**S1 – Example: climate effects on spatial process heterogeneity**

As an example, consider the interception and unsaturated root-zone storage processes in energy limited cool and humid environment. In such environments, large parts of catchments do often exhibit only limited storage deficits and can remain hydrologically connected for much of the year. The elevated precipitation volumes and short inter-storm durations together with limited energy supply for evaporation will result in both stores that are often filled close to their capacity, notwithstanding their potentially significant storage capacities (e.g. forest). As little additional water can be stored, the

systems converges towards a linear response, i.e. what is going in, goes out without significant storage changes and largely independent from spatially heterogeneous storage capacities across the entire catchment. Thus, in that example, any spatial heterogeneity of storage capacities, as for instance dictated by different land cover across the catchment, does not significantly influence the hydrological response and may therefore neither be meaningfully identified by the available data nor actually necessary to account for in a model. As the same applies for other processes, it can be argued that lumped top-

down models, if rigorously tested, may indeed be capable of meaningfully reproducing the observed hydrological response under these specific environmental conditions. However, the more arid the climate and the higher the seasonality of precipitation, the more pronounced the importance of the storage capacities and their spatial heterogeneity become: after a

dry period, forested hillslopes with higher interception and root zone storage capacities than grasslands will need more water to overcome the storage deficits. Thus grasslands will, due to the lower storage deficit that needs to be overcome, establish hydrological connectivity earlier than forests, which has, depending on the areal proportion of the two landscape elements within the catchment, considerable potential to influence the entire catchment response. A lumped formulation of the process

will then indeed lead to a considerable misrepresentation of the hydrological system if a model customized for cool and humid conditions is applied under drier and warmer conditions, and further exacerbated by pronounced differences in topographic relief and/or land cover within the catchment.

**S2 – Example process complexity: Interception**

As an example consider which individual processes a description of vegetation interception at different hierarchal levels of

detail may, amongst others, involve. At the level of individual tree branches, it can be split in into the individual respective interception capacities of a branch and its leaves. While the first is controlled by the mechanical water loading capacity of the branch, which in turn is a function of branch geometry, wind speed, wind direction and precipitation phase, the latter also depends on the phenology of the plant under consideration. Applying classical mechanics, information on material properties and geometry of the branch-leave system together with time series of wind speed, wind direction, energy supply and

precipitation then allows to compute time series of water storage in as well as drip and evaporation from the branch-leave system. At a higher hierarchal level, the level of the individual plant, the detailed, mechanistic description has to be extended to a three-dimensional cascade of individual, interacting branch-leave systems, each characterized by its own position and geometry and therefore affected by differences in wind exposure, direct precipitation input as well as throughfall from systems above. For individual young plants with a few branch-leave systems, depending on how many of the material and

geometric can be determined with some level of confidence, and how many may require some degree of lumping and simplifying conceptualizations, a mechanistic description may remain a feasible option. Yet, the overall interception at the level of a plant is the result of a distribution of different individual thresholds, i.e. interception capacities. With increasing complexity, the resulting non-linear system then becomes increasingly problematic to predict with a detailed, small-scale description, due to uncertainties in boundary conditions, forcing and system states (e.g. Zehe et al., 2007). At the subsequent

stand level, the detailed properties of different plants of the same species but also other species and the composition of plants at a given stand need to be known in addition if interception wants to be treated in a detailed way based on small scale physics. This is effectively not possible with current day observational and computational technology and may for a long time not be. In absence of the required detailed observations, observations at a higher hierarchal level and/or calibration are required to establish a meaningful process parameterization. Both dictate a lower degree of process detail and thus a higher

degree of integration to limit the effects of equifinality. In a system that is set in the realm of organized complexity - too random, i.e. unobservable, to be treated in a deterministic way and too organized to be treated in an exclusively statistical way (Dooge, 1986) - zooming out then often results in the emergence of simple, generalizable functional relationships of the

process under examination (here: interception) with some system properties (here for example Leaf Area Index, e.g. Samaniego et al., 2010) at that scale.

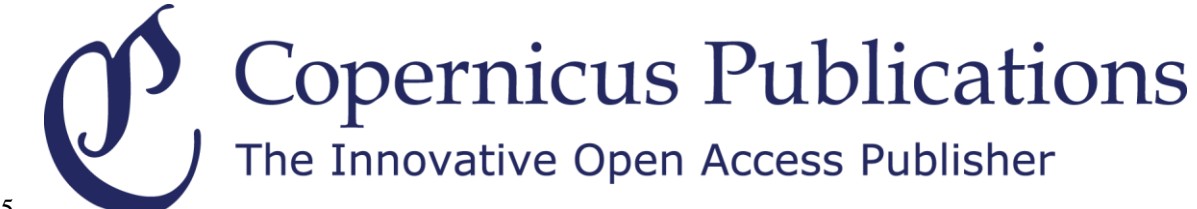