# Peer review of "HESS Opinions: The complementary merits of top-down and bottom-up modelling philosophies in hydrology"

_Hydrology and Earth System Sciences, 2017_

## Referee Comment (RC1) · H. Gupta (Referee) · 2 Feb 2017

"*HESS Opinions: The complementary merits of top-down and bottom-up modelling philosophies in hydrology*" by Markus Hrachowitz and Martyn Clark

Reviewed by Hoshin V Gupta, Professor, The University of Arizona

Feb 1, 2017

**Summary:** Hydrological models exist on a scale of process complexity, from "bottom-up" (BU; often called physically-based) to "top-down" (TD; often called conceptual). The goal of this opinion paper is to improve discussion/exchange between practitioners of these approaches (which are often perceived to be in competition), by a) identifying, discussing and clarifying some common misunderstandings and misinterpretations, b) examining the failures and successes of each approach, and c) highlighting the complementary nature and value of micro-scale process understanding versus the quest for general laws at the catchment scale. The authors discuss some common beliefs such as that "*TD models have a poor physical and theoretical basis*", "*TD models are too simplistic and cannot adequately represent natural heterogeneity*", and "*Top-down models are ad-hoc formulations of untestable hypotheses and always need calibration*".

***Based on this discussion one may conclude (here, I am adding my own perspective to the authors statements):***

a) There needs to be better recognition of the fact that all hydrological models are to varying extents a blend of "*conceptual*" and "*physically-based*" with each approaching the modeling problem from a different perspective – macroscale versus microscale. With this in mind the authors make the reasonable statement that "*an ideal model would provide an equally good representation of both aspects*".

b) There are at least two major functional behaviors that any model must reflect, which are in regards to how water fluxes are partitioned at the near surface and in the unsaturated root zone. This helps to establish the minimal degree of process complexity that must be represented.

c) Because TD models can, in principle, be implemented with any desired process and spatial detail (depending on availability of data), the TD and BU approaches should really be considered to be complementary, and dependent on the availability of sufficient observations at the selected modeling scale and resolution.

d) Neither approach (TD and BU) can escape the need for rigorous, calibration, testing and post-calibration evaluation, because (i) each approach leads to a hypothesis about the nature of the underlying system, and (ii) direct observations of effective parameters at the relevant modeling scale and resolution are typically not available. In this regard, one must be cautious of using and parameterizing models in such a way as to provide deceptive impression of accuracy (particularly true for the BU approach).

e) It would be desirable to establish a mainstream culture of rigorous testing of alternative model formulations (hypotheses), robust model calibration and, and systematic assessment of model uncertainties, based on the establishment of some "*minimum requirements*".

f) Progress in catchment-scale understanding of hydrological functioning and the related development of models for more reliable predictions will be well served by (i) a better understanding of how natural heterogeneities at all scales aggregate to larger scales and how this influences the hydrological response, and (ii) a convergence of TD and BU strategies, in particular with respect to exploiting the features of organization in these complex systems in a hierarchical way. A potential way forward is via large sample, comparative hydrology to identify patterns and generally applicable, functional relationships.

**My Review Comments:** I found little in the substance of this opinion paper to disagree with. My main comments have, therefore, to do with the fact that the presentation tends (I suspect partly unintentionally) to come across as a defense of the TD approach, rather than a balanced evaluation of the strengths and weakness, and complementary nature, of the TD and BU approaches. Certainly in the Gupta et al (WRR 2012, Model Structural Adequacy) paper, of which Clark is a co-author, we argued for the commonality of underlying structure of most if not all hydrological models based on the steps involved in model building, and the need for more cross-fertilization across the modeling community. I very much like the fact that the authors of this paper emphasize the issues of the perceived (but unnecessary) conflict between the TD and BU approaches, but I feel that the argument could be refined and made more balanced by taking note of the fact that many of the points raised in defense of TD modeling are really more general comments that apply to all levels of model complexity – from BU to TD, and revising many of the concluding comments appropriately.

Below, I provide the summary I prepared (of major points presented) while reviewing the paper. While doing so, I found myself generalizing some of the comments made to extend to both TD and BU modeling, and slightly reorganizing the concluding comments. I provide them here in case it helps the authors to see these remarks from a slightly different perspective, and thereby to be useful in strengthening the paper.

In conclusion, I commend the authors on a very nice commentary ☺.

**Summary of the Paper:**

**The issue:**

- Hydrological models frequently fail to reproduce the hydrological response in periods they have not been calibrated for, thereby providing unreliable predictions – this suggests that some of the underlying processes that control how water and energy are stored in, transferred through, and released are not sufficiently well represented.
- Such models exist on a scale of process complexity, from "bottom-up" (BU; or physically-based) to "top-down" (TD; or conceptual).
  - Bottom-up refers to detailed, high-resolution descriptions of small-scale processes numerically integrated to larger (spatial) scales, that include detailed and explicit treatment of conservation of mass, energy and momentum, and whose parameterizations are based directly on observations of fluxes on the small scale.
  - Top-down refers to less detailed representations, often spatially lumped at catchment-scale, whose treatment of conservation and parameterizations are less detailed.

- Unfortunately, there is little fruitful exchange between BU and TD modeling communities, and communication is often limited to mutually highlighting each other's deficiencies. To achieve progress it is important that we:
  - Examine the failures and successes of each approach
  - Appreciate the complementary nature and value of micro-scale process understanding versus the quest for general laws at the catchment scale
- Goal of this paper:
  - To identify, discuss and clarify common misunderstandings and misinterpretations of competing modeling approaches
  - To provide a perspective of how to take advantage of different modeling philosophies so as to improve predictions

**Modelling philosophies:**

- The BU strategy has two main features; (1) Explicitly accounts for spatial heterogeneity, and (2) Provides a rigorous and physically consistent way to encapsulate and formalize theoretical knowledge of dominant processes. In principle, this:
  - Enables hypotheses that can be individually scrutinized and tested against observations
  - Provides meaningful representation of natural feedbacks between individual parts of the system -- implementations of this inductive approach to hydrology have the potential to reproduce emergent patterns.
- The TD strategy is based on representing emergent, _system-integrated_ (catchment scale), response patterns reflected by information in available data (mainly limited to areal estimates of precipitation, potential evaporation and stream flow). It aims to reproduce the observed dispersion patterns that reflect catchment-internal organization and feedback. While maintaining mass balance and parsimonious representation of the energy balance, it does not explicitly define and describe the detail of underlying processes.

**Discussion of Typical Modelling Beliefs:**

- **Top-down models have a poor physical and theoretical basis**
  - While having an element of truth, it is demonstrably clear that observation-based, functional relationships at the macroscale can be valuable descriptors at the macroscale without loss of essential information.
  - Since water flows in a catchment follow the observable, physical phenomenon of dispersion, controlled by water and energy input, gravity, and flow resistances, TD model development is the attempt to identify functional input-output relationships between that emerging through organization at the catchment scale.
  - Accordingly, it should be possible to test competing hypotheses without resorting to small-scale physics. By examining a large enough sample of systems, the emerging patterns and associated (tested) functional relationships can facilitate similarity analysis and classification, and facilitate the "_search for general laws at the macroscale_".
  - From this perspective, TD models can be considered as "physically-based" and parsimonious representations at the macroscale. Given that they consist of large-scale conservation equations, they necessarily require large-scale flux

parameterizations (i.e. the closure problem) that act on system-average water quantities. The challenge is to assign clear physical meaning.

- o We argue, therefore, that well implemented and tested TD models can have a robust physical and theoretical basis, and that it is possible to relate the structure of top-down models to stores and fluxes in nature, albeit at a different spatial scale and process resolution than bottom-up models.
- o There is, however, room for improvement, such as (a) providing an explicit physically consistent treatment of energy and momentum balances, and (b) determining the level of detail at which dominant catchment processes must be resolved to reproduce the observed system response in a meaningful way.

- **Top-down models are too simplistic and cannot adequately represent natural heterogeneity**
  - o It is true that TD models are often assumed, a priori, to be meaningful representations of the system, rather than treated as hypotheses to be tested. While displaying high skill with regard to a calibration objective, they can struggle to reproduce additional system internal dynamics and emerging patterns, resulting in poor predictive power.
  - o While lack of adequate model calibration, testing and evaluation partly arises both from (a) insufficient exploitation of information in available data, and (b) lack of suitable data to more effectively constrain models, many models remain ill-posed inverse problems.
  - o Two issues are of importance here – Process and Spatial complexity:
    - *Process Complexity:* All models of terrestrial hydrological systems must represent the two major types of partitioning that control how water is stored in and released through upward, downward or lateral fluxes. The first is the near-surface partitioning of precipitation into (i) evaporation and sublimation, (ii) overland flow and (iii) infiltration. The second is the partitioning of infiltrated water into (iv) soil evaporation, (v) plant transpiration, (vi) shallow, lateral subsurface flow, and (vii) percolation. Regardless of (TD or BU) strategy, the level of detail for resolving individual processes at the two partitioning points must be decided. Questions to be answered are: *How much detail is necessary to reproduce observed dynamics and pattern? How much detail is warranted by the available data to meaningfully parameterize and test the chosen process representation?* As long as simplification encapsulates relevant dynamics of the system, lumping does not necessarily involve loss of information, and has the potential to provide a description that is consistent with real world observations at the scale of interest.
    - *Spatial complexity:* The degree of spatial complexity that can be incorporated in a model depends on the detail of available information (which types of heterogeneity are present, how do they affect water storage and release, and which can be captured by a single emergent functional relationship?). In principle, TD models can be formulated at any level of process and spatial complexity, limited only by available information. So the decision regarding which level of detailed process representation is feasible/necessary must be made by the modeler on basis of the available observations. It is also important to ask whether BU models may actually be deceptive about the accuracy that is

implied by their formulation, when appropriate observations at the scale and resolution of interest are not available.

- **Top-down models are ad-hoc formulations of untestable hypotheses and always need calibration**
  - o It is true that applications of TD models often assume their appropriateness rather than being treated as hypotheses to be tested. More correctly, different environmental conditions should dictate the specification of prior parameter information and/or process inclusion/activation.
  - o Further, _all_ models (TD or BU) are aggregations of distinct hypotheses that should ideally be tested independently to avoid the adverse effects of equifinality. The question is "*To which level do model components then have to be disaggregated to be constitute testable hypotheses*?" Inevitably, any treatment of any model as a single hypothesis is likely to remain a weak test. In contrast, individual testing of system sub-components provides more information because the sub-components are necessarily less complex than the overall model and results in a stricter test.
  - o So this is not a limitation specific to TU models, and the main challenge is availability of relevant observations at the scale and/or resolution of interest, which gives rise to the need for calibration. Meanwhile, weak model tests are preferable to no tests at all.

**Implications, potential ways forward and concluding remarks:**

- All hydrological models are to some extent "conceptual" and to some extent "physical"; they largely only differ in the degree of detail they resolve the system, which in turn is dictated by the available data. TD models approach the problem from a macroscale understanding, while BU models emphasize the micro-scale perspective. Needless to say, an ideal model would provide an equally good representation of both aspects.
- TD and BU modeling are complementary, with TD modeling constituting a physical-based approach to hydrological modeling that can be firmly rooted in holistic empiricism, similar to statistical physics.
- TD models can, in principle, be implemented with any desired process and spatial detail (depending on availability of data).
- TD and BU models must both reflect our conceptual understanding of the system in regards to how water fluxes are partitioned at the near surface and in the unsaturated root zone. Since all the relevant fluxes can be present in any environment (with different relative importance), all TD top-down models must have the same fundamental model structure (but not necessarily the same parameterization) to reflect these processes.
- In the absence of sufficient observations at the modeling scale and resolution, all hydrological models (TD and BU) remain hypotheses and require rigorous, testing and post-calibration evaluation. The fundamental problems in catchment modeling do not lie in the type of model used, but rather in the way a model is applied.
- Progress in catchment-scale understanding of hydrological functioning and the related development of models for more reliable predictions hinges on a better understanding of how natural heterogeneities at all scales aggregate to larger scales and how this influences the hydrological response. Although acknowledged in the early 1980s,

remarkably little progress was made. A potential way forward is large sample, comparative hydrology to identify pattern and generally applicable, functional relationships.

- All hydrological models (TD and BU) applied at scales beyond the plot scale require some degree of calibration, as direct observations of effective parameters at these modeling scales and resolutions are typically not available. Improper application of BU model parameters from observations that do not match the modeling scale and/or resolution can lead to deceptive impression of accuracy.

- If both TD and BU models are to be useful as scientific tools, a mainstream culture of robust model calibration (and post-calibration), rigorous testing of alternative model formulations (i.e. hypotheses), and systematic assessment of model uncertainties (including parameterization) must be achieved. This should involve evaluation with respect to multiple variables and model states, including (data permitting) model sub-components, as well as to multiple criteria, and the establishment of "minimum requirements".

- Taken together, these arguments suggest that somewhat arbitrary dichotomy between TD and BU models tends to lead to some degree of confusion. In reality, all models are to some degree conceptual. The real issues have to do with:
  o The inadequate ways in which models are applied. Recognizing this may help shift the focus towards the real fundamental questions in catchment-scale modeling, including "how much detail do we need in our models?" and "how much detail is warranted by data?"
  o Finding a balance that allows us to best describe the system based on scientifically robust grounds
  o Accepting a more rigorous culture of model testing, to adjust process and spatial complexity to the environmental conditions and data availability in specific catchments, so as to reduce the risk for oversimplifications and system misrepresentations (TD) while embracing the value of zooming out and making use of emergent processes (BU).

- On balance, we believe that improved hydrological understanding will require a convergence of TD and BU strategies, in particular with respect to exploiting the features of organization in these complex systems in a hierarchical way. We strongly encourage researchers to both acknowledge and actively use the advantages of both modeling strategies in order to strengthen the science of hydrology.

---

## Referee Comment (RC2) · R. Loritz (Referee) · 10 Feb 2017

**Review of "HESS Opinions: The complementary merits of top-down and bottom-up modelling philosophies in hydrology"**

**By Markus Hrachowitz and Martyn Clark**

**Summary, recommendation and general comments:**

In this manuscript the authors share their opinion about the ongoing discussion on different competing modelling philosophies in hydrology. Besides describing the different modelling approaches in detail, they also discuss what they call "various modelling myths". In the end they propose a way forward and give some recommendations for hydrological modelers.

I do agree with the authors that the discussion about the different modelling philosophies is sometimes rather driven by emotions than by facts. I also think that an opinion on this issue and proposals for a way forward could be of interest for publication in HESS. However, I believe that before this paper can be accepted for publication substantial revisions are needed.

First of all, both authors have a separate opinion paper or comment with a closely related content in HESSD at the moment (Clark et al., 2017; Savenije and Hrachowitz, 2016). Especially the discussion and the review of the opinion paper by Savenije and Hrachowitz (2016) cover a lot of similar points and arguments as this paper. But also the comment by Clark et al. (2017) has several overlapping arguments, especially related to the propsal about how to progress in hydrological modelling. With three papers in HESSD covering similar topics I think it is especially important that the authors clearly show what this opinion paper differentiates it from the other two manuscripts.

My second concern is that a substantial part of this paper reads like a text book. While the language is clear and easy to follow, I was wondering if the potential audience really needs a two page long introduction to "conceptual" and "physically-based" models? Similarly, other sections seem to be redundant as they have already been covered in great detail in several opinion, comment and review papers (e.g. Bahremand, 2015; Clark et al., 2015; Gupta et al., 2012).

This brings me to a more general comment aimed at all opinion papers which is that careful reading is required to identify where facts end and the opinion of the authors starts. One example for this paper is when the authors write that top-down models have "a parsimonious representation of the energy balance". Is this a fact and has it been shown somewhere or is this an opinion? As far as I know, most hydrological models do not close the energy balance or even keep track of the energy in the system. How can you know if you close the energy balance, when you only try to close the mass balance?

Another example is the unclear separation of the macro- and microscale in this paper. For instance the authors argue that macroscale models are important and physically-based with e.g. Sivapalan's (2005) search for a general law at the macroscale or with a comparison with Gay Lussac's law. However, the papers they mention to support this argument use often macroscale models to define various states of the microscale, for example the root zone storage. While using macroscale models to estimate states at the microscale is a perfectly valid approach, it is very important to make clear to the reader that this can

only be an estimate and is rather difficult because of the high degrees of freedom we have in hydrology. A precise definition of the macro- and microscale and a clear structure of the manuscript in this context might help to improve this paper and would ensure that not even more "modelling myths" are generated.

As I have the highest respect for both authors I am sure they have great and positive ideas for the future of both modelling philosophies. However, I believe that we do not need another paper where we discuss how physically-based or not the different modelling philosophies are. I recommend that you focus on the complementary merits of both approaches. Furthermore, I suggest giving clear examples and sharing your ideas how we could for instance combine top-down and bottom-up models in practice. This could make the manuscript much more unique and meaningful. As I believe that the discussion of this topic is of relevance for the hydrological community I hope my comments, questions and opinions are constructive and can help to improve this manuscript.

**Comments, corrections and questions:**

**Section 1 What is the issue:**

*Page 2 Line 28:* Maybe add some references where the authors showed that their model failed after the calibration period, both from the bottom up and top down community.

*Page 3 Line 6:* Could you define catchment scale?

*Page 3 Line 10-11:* What do you mean here with "respect to bottom up models".

*Page 3 Line 11 – 12:* I couldn't find the part where you provide a perspective of how to take advantage of different modelling philosophies.

**Section 2 Modelling philosophy:**

This section is mostly written clearly and precisely. Nevertheless, I think the potential reader of this opinion paper is already familiar with the different modelling approaches and reading this section is very akin to reading a text book. I would consider shortening this section with references to other studies or textbooks.

*Page 4 Line 4 – 5:* From my point of view the scenario in which you end up in a catchment where you only have reliable runoff and rainfall data but nothing more available is rather unrealistic: In which catchment in the world do you have reliable streamflow, evapotranspiration and rainfall measurement but no other information of the catchment? At least in Europe and the US you have land cover and geological maps. Furthermore, if there is a gauging station and rainfall measurements, most likely a person is doing maintenance on the respective instruments on a regular basis. This person will most likely accumulate a lot of qualitative information about the hydrological functioning of the catchment and could possibly also complement this picture with low-effort additional measurements or soil sampling. For instance Jackisch et al. (2014) showed how fast one can characterize a remote meso-scale catchment based on a brief measurement campaign. If land cover is managed forest or agriculture,

frequently nationwide reports on productivity and for example drought risks are available. We have digital elevation models for the whole earth in decent resolution, monthly estimates of precipitation and soil moisture from satellites and so on. In my opinion the problem is often very different from the projected scenario: We do not know how to use the data in our hydrological models or if it is of relevance. But I admit that this may be a different story.

*Page 4 Line 5-8:* Is the "system integrated response pattern" really the "starting point" of top-down models? Isn't the starting point the delineation of a catchment based on the surface topography assuming a closed water balance? Since most top-down models are calibrated on the streamflow, do you mean streamflow by the term "system integrated response pattern"? Consider clarifying what you mean with the terms, maybe some examples beyond stream flow, and what you mean with "starting point" here.

*Page 5 Line 12:* Could you please explain in more detail what you mean with a parsimonious representation of the energy balance?

**Section 3 Modelling myths**

*(C1) "Top-down models have a poor physical and theoretical basis"*

*Comparison with Gay-Lussac's law:*

I think that the comparison with Gay-Lussac's law and the top-down modelling approach is a little misleading. I am not saying top-down models are not physically based. Like most hydrologists I believe that this entire discussion is based on an ill-posed definition and classification of hydrological models into the dichotomy of physically-based and conceptual models. However, with Gay-Lussac's law you can describe the macroscopic state of a system. But you can't say anything about the microscopic state of the system, for example where the molecules really are. Following your arguments and speaking of top-down models now this would mean that you can't say anything about the microscale of a catchment, for example where the water is in your catchment. However, later you argue that you can identify the root zone storage with a top-down model. Is this not part of the microscale? With a macroscopic model you can only infer about the microscale if you constrain the possibilities of the microscale using either additional measurements or process-based reasoning with the help of statistics. However, this is really difficult in hydrology due to the large number of degrees of freedom. For example, if your model is calibrated to mimic the runoff generation and if we assume for a second that the two water worlds proposed by McDonnell (2014) are real, there is no information about the root zone storage in the rainfall-runoff data and it is really difficult to know if what you learn from your models is true.

Overall, it is not clear where you want to go here. A top down model is based on 1.) the conservation of mass and 2.) on the delineation of the landscape into some kind of control volumes mostly in form of a catchment. With a top down model you can hence make assumptions about the macrostate of a catchment or of a similar control volume. With the help of statistics, process-based understanding or additional measurements you might be able to get a grasp of the microscale. So why are you comparing it with a natural law which is constrained by the energy and mass conservation when the model you defend is not? I believe most hydrologists know how a conceptual model works so is this whole

comparison necessary at all? Maybe a rigorous definition of macroscale and microscale might help to improve and clarify differences, similarities and linkages between top-down and bottom-up models?

*Page 6 Line 3:* The molecular dynamics approach might be untestable and unfeasible but certainly not unnecessary. It is the theoretical basis of the movement of gas particles and hence necessary if you want to understand a system.

*Page 6 Line 23*: Can you please explain in more detail what you mean with parsimonious representation of the energy balance, again?

*Page 8 Line 1:* Holistic empiricism and on *Page 7 Line 6* assign physical meaning to them a priori? Please explain why the two statements are not in contradiction.

*(C2) "Top down models are too simplistic…" and (C3) "Top-down models are ad-hoc formulations…"*

Both sections are written clearly and well but I think this has all been said and written down several times. You might consider to shorten this section.

*Subsubsection 3.2.2 and 3.2.1:* What do you mean with process and spatial complexity. Could you please define complexity and how it relates to the respective models?

Page 10 Line 30-31: Is it really multivariate observed response dynamic? At least in one of the cited examples the authors only use streamflow and derivations of it.

**Section 4 Implications, potential ways forward and concluding remarks**

*Page 12 Line 19 - 20:* "Competing approaches" Despite the title of the manuscript I had the feeling that the main focus was on defending top-down models. Why do you stress the dichotomy although I understood the overall aim of your opinion paper to be exactly the opposite?

*Page 13/14 Line 34 / 1-2:* I think this sentence is a little misleading. Obviously you can use a Darcy-Richards based model on the macroscale. However, you need to use a rather fine discretization of the model elements.

*Page 14 Line 16 - 17*: Why are you so pessimistic here? Maybe you could add some references so the reader can better understand your pessimism.

*Page 15 Line 9*: If this is about hydrological modelling at the catchment scale you might consider adding catchment scale modelling to the title. Unfortunately it is not clear what the catchment scale is. It would be nice if you could add a definition.

References

Bahremand, A., 2015. HESS Opinions: Advocating process modeling and de-emphasizing parameter estimation. Hydrol. Earth Syst. Sci. Discuss. 12, 12377–12393. doi:10.5194/hessd-12-12377-2015

Clark, M.P., Bierkens, M.F.P., Samaniego, L., Woods, R.A., Uijenhoet, R., Bennet, K.E., Pauwels, V.R.N., Cai, X., Wood, A.W., Peters-Lidard, C.D., 2017. The evolution of process-based hydrologic models: Historical challenges and the collective quest for physical realism. Hydrol. Earth Syst. Sci. Discuss. 1–14. doi:10.5194/hess-2016-693

Clark, M.P., Fan, Y., Lawrence, D.M., Adam, J.C., Bolster, D., Gochis, D.J., Hooper, R.P., Kumar, M., Leung, L.R., Mackay, D.S., Maxwell, R.M., Shen, C., Swenson, S.C., Zeng, X., 2015. Improving the representation of hydrologic processes in Earth System Models. Water Resour. Res. 51, 5929–5956. doi:10.1002/2015WR017096

Gupta, H. V., Clark, M.P., Vrugt, J. a., Abramowitz, G., Ye, M., 2012. Towards a comprehensive assessment of model structural adequacy. Water Resour. Res. 48, 1–16. doi:10.1029/2011WR011044

Jackisch, C., Zehe, E., Samaniego, L., Singh, A.K., 2014. An experiment to gauge an ungauged catchment: rapid data assessment and eco-hydrological modelling in a data-scarce rural catchment. Hydrol. Sci. J. 59, 2103–2125. doi:10.1080/02626667.2013.870662

McDonnell, J.J., 2014. The two water worlds hypothesis: ecohydrological separation of water between streams and trees? Wiley Interdiscip. Rev. Water 1, n/a-n/a. doi:10.1002/wat2.1027

Savenije, H.H.G., Hrachowitz, M., 2016. Opinion paper: How to make our models more physically-based. Hydrol. Earth Syst. Sci. Discuss. 0, 1–23. doi:10.5194/hess-2016-433

Sivapalan, M., 2005. Pattern, process and function: elements of a unified theory of hydrology at the catchment scale. Encycl. Hydrol. Sci.

---

## Referee Comment (RC3) · T. Wagener (Referee) · 13 Feb 2017

The authors, as always in their papers, have written a well-formulated discussion of relevant current issues in hydrological modeling. While there are many interesting points here, and Hoshin points out quite a few, I have to agree with Ralf Loritz's comment that it becomes hard to keep track of what they key points are in an increasing number of commentaries on (at least seemingly) similar issues. In the case of the present manuscript, I think that there are some issues that can be discussed with more rigour to highlight its uniqueness (though the authors might disagree).

One thing that stands out in this commentary is the explicit use of the term top-down modeling. It is not clear to me though what definition the authors use for top-down

modeling. My understanding of the manuscript suggests that here this definition includes all conceptual type approaches to hydrologic modeling. So, are all conceptual modeling approaches equivalent to a top-down modeling philosophy? I do not think so, though the authors likely have a different point of view (which would be fine). What definition do the authors follow? Is this defined by the model type I use (ODE vs PDE) or by the mindset/objective I have when developing my model?

Following some of the early definition top-down modeling "provides a systematic framework to learning from data, including the testing of hypotheses at every step of analysis" (Sivapalan et al., 2003). This is often applied in a hierarchical manner (e.g. using signatures), but not necessarily so. If this is the definition the authors use, then I do not think that models such as HBV have been developed following a top-down modeling philosophy. They rather have been developed with a bottom-up mindset I think. Similarly the Sacramento model was not build to just fit the data, but based on an attempt to provide a simple representation of physics. Is there really a common philosophy underlying the modeling approaches used to build HBV, in the top-down papers by Sivapalan and colleagues, and in the FUSE framework? Is it really a binary decision whether an approach is top-down or bottom-up?

If I assume that the definition by Sivapalan above is appropriate, then some important contributions to top-down modeling are missing from this paper. Most notably is the work by Peter Young (e.g. Young, 2003 and much earlier than that), who, with his data-based mechanistic approach, has provided one of the few very structured frameworks for top-down modeling. Of course he did so by making some strong assumptions, which limit the generality of his approach. It would be good if the authors could have a wider look at literature in which top-down modeling strategies are investigated (if they use the term more narrowly than simply all conceptual models).

I think by using a very wide definition of top-down modeling, we miss the opportunity to discuss some important remaining problems. Mainly that hydrology still lacks "a systematic approach to learning from data" as proposed by Siva. For example, how do

we assess model complexity (given that information criteria typically do not work for hydrologic models), so that we can identify the simplest model that fits the data? How do we decide that one model structure is better than another one beyond just looking at performance? The data-based mechanistic approach provides a nice strategy to identify the simplest representation (of routing) supported by the data, while also allowing for a hydrological interpretation. I do not think that we have a more general framework of this type yet (i.e. without Peter's assumption of using linear transfer functions etc.).

I am also unclear why a top-down approach should be restricted to catchment scale observations (if that is what the authors suggest). If the approach is focused on learning from data then its philosophy can be applied at any scale. Work by Young and colleagues using their top-down philosophy have not been limited to catchment scale hydrologic data, so why should it be for us in hydrology? We could actually build distributed models using a top-down strategy for catchments with extensive internal observations.

These are just some thoughts to (hopefully) advance the discussion.

References

Sivapalan et al. (2003). Downward approach to hydrological prediction. Hydrological Processes, 17, 2101-2111.

Young, P. (2003), Top-down and data-based mechanistic modelling of rainfall-flow dynamics at the catchment scale, Hydrological Processes, 17, 2195-2217.

---

## Referee Comment (RC4) · M. Bierkens (Referee) · 14 Feb 2017

I started to read this opinion paper with great anticipation because I think there is a desperate need for joining top-down and bottom-up approaches to arrive at solid hydrological theories. The paper is generally well written and starts out with a promising small review about the nature of bottom-up and top-down approaches.

However, after reading the part thereafter, I have to admit I started to become a bit disappointed. The reason for this is that the second part of the paper becomes quite unbalanced and reads as an apologia for top-down modelling. What I miss is a section "Modelling myths or not" for bottom-up approaches. For example, statements as "Bottom up models are over-parameterized" can be elaborated on. After that I would

have liked to have a section to sketch a way forward to marry both approaches taking account of their complementarities. Shortening the "Modelling myths or not" to make room for similar sections on bottom-up approaches would make the paper much more balanced and interesting.

Also, I have some additional remarks about the paper.

First, the authors underpin the statement that "At the macroscale, which in the realm of organized complexity is frequently characterized by the emergence of relatively simple functional relationships. . . that integrate typically unobservable natural heterogeneity over the model domain", with a comparison with to statistical physics (e.g. gas laws). However, there is a big difference between an ideal gas and a hydrological system related to the assumption of ergodicity. In that context, this assumption loosely means that at all times all microstates are present when averaging over the volume. This assumption is valid for an ideal gas but not necessarily the case for hydrologic systems.

Second, I feel that a problem with the way top-down megascopic hydrological laws are derived (also in comparative hydrology) is that often only (signatures) of the output variables are used to assess the form of the $Q = F(S)$ relationship. This can only be done if a certain form (often a power function) is assumed a priori. I think that to really assess the form of these relationships one needs to jointly measure the state (groundwater storage, soil moisture, snow water equivalent) and the output variables (discharge, evaporation). Very rarely these state observations are used or available in catchments used in comparative hydrology. So we should get away from the fixation with hydrographs only and start measuring states. To add to this: energy conservation is often added by checking if the found megascopic laws follow Budyko's hypothesis. This is only a weak check on energy conservation, because it only checks for very long times and doesn't guarantee energy conservation at any given time.

Third, once megascopic laws have been derived empirically, these laws' physical basis should be strengthened by also deriving them from upscaling from smaller-scale me-

chanics. A well-known example is Darcy's law. It was first established empirically - note that this was done by both observing states (heads or actually the head gradient) and fluxes. Later (much later), it was shown that it could be derived from the Navier-Stokes equations (by 1. neglecting quadratic inertia terms: laminar flow -> Stokes equations; 2. volume averaging by homogenization; 3. noting that drag forces are much larger than viscous forces). Obviously, heterogeneities in hillslopes and catchments are more complex than pore-scale heterogeneities in a REV. This makes simple homogenization not likely a suitable approach. However, hyper-resolution (cm-scale) modelling using simulated heterogeneities (including macropres etc) with 3D PDE-based models (e.g. Parflow, Hydrogeopshere, Cathy) and upscaling the results may be a way to derive megascopic laws from first principles.

---

## Short Comment (SC1) · 14 Feb 2017

Title: HESS Opinions: The complementary merits of top-down and bottom-up modelling philosophies in hydrology

Authors: Markus Hrachowitz, Martyn Clark Journal: Hydrology and Earth System Sciences

Review:

Hydrological models are used to predict floods, droughts, groundwater recharge and land-atmosphere exchange, and are of critical importance as tools to develop strategies for water resources planning and management. In these hydrological models, two
modelling philosophies, namely bottom-up and top-down approaches are the basis in representing a hydrological system. In bottom-up approaches, very detailed representations of the hydrological system is considered. On the other hand, in top-down approaches, less detailed, often spatially lumped representations of the hydrological system is considered. As underscored in the current literature, it has been extensively argued in numerous journal papers about the pros and cons of top-down and bottom-up approaches.

In this manuscript, the authors scrutinize common modelling critiques on top-down approaches and discuss the extent to which they are justified.

Based on this review, the following comments are made:

1) The current version of the paper does not convince that the cited papers are sufficient and informative for the authors to draw conclusions or comments on the topic that is discussed in this paper. Moreover, from the reader's point of view, what has been discussed in this paper has already been echoed in the current literature.

2) It has been extensively argued in numerous journal papers about the pros and cons of top-down and bottom-up approach. Therefore, from the reader's point of view, for this commentary to have some merits, the authors need to go beyond what has been understood in the current literature. From the reader's point of view, it would be more useful, for example, if the authors bring the concept of middleware that lies in between the said approaches of modeling (i.e., top-down and bottom-up).

3) In the current version of the paper, the authors scrutinize common modelling critiques (C1-C3). Are these critiques developed by the authors? Are these critiques developed based on some published survey? What motivated the authors to consider these critiques as the "common" modelling critiques?

4) In the current version of the paper, the authors scrutinize common modelling critiques on top-down models (C1-C3) and discuss the extent to which they are justified.

[Figure]

From the reader's point of view, the title of the paper does not fit the content of the paper.

5) Referring to line number 22 on page number one, the authors state that the models frequently fail to reproduce the hydrological response in periods they have not been calibrated for, thereby providing unreliable predictions. From the reader's point of view, this statement needs to be cited.

6) In the current version of the paper, the authors discuss about the spatial complexity, process complexity, and spatial scale. However, referring to line number 22 on page number one, from the reader's point of view, it would be more useful if the authors discuss about the influences of temporal scale and its complexity on the said approaches of modeling (i.e., top-down and bottom-up). Is it scientifically justifiable that the processes that are modeled at a particular temporal scale do not change when the temporal scale changes? In the current literature and the modeling practices, the processes that are modeled are the same regardless of the temporal scale of the simulation.

7) Referring to line number ten on page number one, a better understanding bears the potential of identifying the complementary value of the two philosophies for improving "our" models. Are these models developed by the authors? Is this commentary about the models developed by the authors?

8) From the reader's point of view, some of the paragraphs are repetitive (e.g., the paragraphs about the activation and deactivation of processes).

http://research.abzwater.com/review/ABZR6.pdf

---

## Author Response (AR1)

Dear Editor, dear Prof. Zehe,

We thank you for your positive assessment of our manuscript and the detailed, interesting and highly constructive additional comments you provided.

Please find below our replies to these comments, which were, together with the comments of the other reviewers also incorporated into the revised version of our manuscript. Briefly, large parts of the manuscript were re-structured and the major changes include (1) a more rigorous distinction between alternative modelling strategies, (2) a more balanced discussion to better reflect the individual strengths and weaknesses of the different modelling strategies and (3) an additional section, outlining potential ways forward towards a convergence of the different modelling strategies.

**Editor Comments**

Editor comment:
A key for making to approach the issue is maybe to reflect about the complementary merits of these model categories depending on their purpose? As highlighted by Marc Bierkens the merits depend on what we want to model, catchment scale integral response or space time dynamics of state variables. On top of that we might distinguish models for predictions from models for explaining (multi) causal relations in terrestrial system functioning. The sets of useful models for these to paradigms are not necessarily identical and the model paradigm in hydrology is strongly biased towards the prediction issue. This is of course due to the operational origin of hydrological models stream flow predictions.

*Reply:*
*We fully agree and have made this clearer and more explicit in the revised manuscript.*

Editor comment:
In line with reviewer Thorsten Wagener I think that any meaning full model effort combines top down and bottom up thinking, this is not so much a quest of using PDE or ODE. So it might be helpful to better distinguish the models and the approaches to the model problem.

*Reply:*
*Agreed. To allow a clearer and potentially more meaningful distinction between different models we have now included a section "Model Taxonomy", in which we provide a range of different perspectives and argue that a classification framework along a 2-dimensional continuum of spatial resolution and process detail may be more suitable to classify models than the common conceptual-physically based duality.*

Editor comment:
In line with Marc Bierkens I think that a section on modelling myths with "physically" based terms would yield a more balanced story line. Alternatively you could also approach the story by working out the key assumptions underlying both model philosophies (which are pretty different) and to reflect when they become invalid, because this shows the way to progress.

*Reply:*
*We agree and split section 3 into the two contrasting perspectives. In addition we made it more explicit that different models may be suitable for different purposes.*

Editor comment:
For instance with respect to "spatially explicit models": In addition to what has been said by the authors, you might consider to add that spatially explicit models treat fluxes as the product of a driver (potential gradient) and a loss term (conductance). The definition of a potential requires local equilibrium or/ well-mixed conditions in the grid cell. Applications a grid scales larger than 1 m become therefore questionable (even if the land surface model community will not like this point).
The second big assumption in Darcy law is a) a purely diffusive flux which implies no kinetic energy in the flux and b) that gravity driven flux and capillarity driven flux are controlled by the same conductance – also during rainfall driven conditions. Both assumptions might be not appropriate when dealing with fast subsurface flows during rainfall driven conditions.

*Reply:*
*We also fully agree here. This is now made clear in section 3.1.2*

Editor comment:
A key assumption for conceptual models is that catchments as a closed control volume do really exist and that surface water shed provides their boundaries. This assumption can in fact not be rejected within this paradigm, it is almost an axiom. Without relying on that one cannot compile a mass balance by separating of the input (assumed as the total rainfall) and total liquid loss (streamflow) and the residual (equal to storage and evaporation and transpiration). This has the nice implication that runoff generation in conceptual models is a continuous function of storage on a compact interval (between 0 and 1). This implies we can fit a polynomial to this function, due to the Weierstrass theorem. Spatially explicit models do not depend on the validity of the catchment idea.

*Reply:*
*While we agree that conceptual models need to define a control volume, which is mostly done following the surface topography, and that this control volume may be a misrepresentation of reality, we are not sure why this should not be the case for spatially explicit models. Every model control volume (i.e. grid cell) routes water following +/- the steepest gradient and least resistance. The set of grid cell whose water is drained into a specific point than constiotutes the catchment scale control volume. As the subsurface properties are frequently not known in detail, elevation head may be the first source of heterogeneity. As such the surface topography then, in our opinion, also constitutes the first order control on the definition of a catchment.*

Editor comment:
I very much like characterization of catchment as low pass filters and the issue of dispersion. Yet this implies that event convolution (the classical integral model for linear systems) or even peace wise linearization is difficult, as this implies the transfer functions depend or conditional to the input and the state (which implies one can only integrate a short time) and then has to update the kernel. This is why modern conceptual models use time stepping and conceptual representation of the kernel/transfer functions for state updating.

*Reply:*
*We fully agree. This is (for solute transport) nicely shown in the recent work of e.g. Botter et al. (2011) or Rinaldo et al., (2015).*

Editor comment:
As proposed by Mark Bierkens I would also encourage the authors re-think about their macro-microscale argumentation in their response to reviewer Ralf Loritz. I think the key point is that we

cannot infer backwards on the microstate of the terrestrial system (e.g. the pattern of root depths) by knowing the macro state of the system – which implies that many subscale system configurations with purely stochastic and highly structure variability might be represented by the same macro state. This might not bother if we are interested in stream flow predictions, but if matters if we are interested in eco hydrology or distributed state dynamics. Whether the macroscale representation might be favorable or not, depends on our interest – so does the choice of the model, do we want to predict or to explain.

*Reply:*
*Agreed. We have clarified that in the manuscript.*

Editor comment:
I agree with Ralf Loritz that neither physical nor conceptual models close the energy balance (which involves more that the land surface energy balance and soil heat flux, but also the interplay of potential and capillary binding energy of soil water as well as export of kinetic energy in stream flow). I think the this section issue needs to be formulated in a less ambiguous manner

*Reply:*
*Agreed. We have adjusted this description in section 3.1.1.*

Editor comment:
I would encourage the authors to better define how to measure process and spatial complexity, it is the a) the same as predictability (complex processes are more difficult to predict) or b) the amount of information in a time series, c) the number hidden dimensions, d) or is it the degree of non-linearity of the PDE, possible chaotic nature, and state dependent error propagation, c) the number of independent parameters (dimensionality) … I know this is not easy, but I feel that these terms might have many different meanings in the community.

*Reply:*
*Agreed. We have provided a clear definition for our use of the term "complexity" in section 1.*

Editor comment:
Page 1: Intro Beyond the errors arising from uncertain data there might be much from measurement errors, which would become clear if we added error bars at least to our observation data.

*Reply:*
*Agreed and adjusted.*

Editor comment:
Page 5: I am not sure what spatial organization of connectivity means?

*Reply:*
*Sentence now removed*

Editor comment:
Page 5: Top down models are not based on observed input – output relations ship but on their estimates based on input output data. (Otherwise we had no uncertainty if those were observable)

*Reply:*
*We fully agree. Sentence now removed.*

Editor comment:
Page 9: What is actually meant with process complexity – the order of the differential equation and its degree of non-linearity, what is meant with spatial complexity. The degree of spatial detail, isn't this rather information than complexity?

*Reply:*
*We clarified the terminology and proved a clear definition of our use of "complexity" in section 1.*

**Reviewer #1 (Hoshin Gupta)**

Comment:

I found little in the substance of this opinion paper to disagree with. My main comments have, therefore, to do with the fact that the presentation tends (I suspect partly unintentionally) to come across as a defense of the TD approach, rather than a balanced evaluation of the strengths and weakness, and complementary nature, of the TD and BU approaches. Certainly in the Gupta et al (WRR 2012, Model Structural Adequacy) paper, of which Clark is a co-author, we argued for the commonality of underlying structure of most if not all hydrological models based on the steps involved in model building, and the need for more cross- fertilization across the modeling community. I very much like the fact that the authors of this paper emphasize the issues of the perceived (but unnecessary) conflict between the TD and BU approaches, but I feel that the argument could be refined and made more balanced by taking note of the fact that many of the points raised in defense of TD modeling are really more general comments that apply to all levels of model complexity – from BU to TD, and revising many of the concluding comments appropriately.

*Reply:*

*We highly appreciate the reviewer's very positive assessment. After re-analysing the manuscript from the perspective of all reviewers, we agree that it comes across more like a defence of top-down models rather than the intended balanced evaluation of the two modelling strategies. We will accordingly re-structure and re-formulate the relevant sections in the revision.*

*Having said that, and given that also the other reviewers noted that the paper should be less a defence of top-down models, we would also like to stress one, potentially not irrelevant point: bottom-up models, i.e. "physically-based" may largely benefit from a semantic-psychological bias. The term "physical-based" inherently implies that they are "correct" descriptions of real world-systems, which further implies that all other models are not "physical" and thus less "correct". From this perspective, we believe that any type of comparison between bottom-up and top-down strategies will to some extend necessarily come across as a defence of top-down models, i.e. explanations of why they can be as meaningful representations of reality as bottom-up models. In other words, already the term "physically-based" puts bottom-up models in the (often not really justified) position of benchmarks other models have to be compared to, even if they are not necessarily "better" descriptions of reality.*

Comment:

Below, I provide the summary I prepared (of major points presented) while reviewing the paper. While doing so, I found myself generalizing some of the comments made to extend to both TD and BU modeling, and slightly reorganizing the concluding comments. I provide them here in case it helps the authors to see these remarks from a slightly different perspective, and hereby to be useful in strengthening the paper.

*Reply:*

*We thank the reviewer for the reorganization and generalization of the main points. We believe these adaptations will add substantial value to the manuscript and we will adjust the text accordingly.*

Comment:

In conclusion, I commend the authors on a very nice commentary.

*Reply:*

*We thank the reviewer very much for this encouraging assessment!*

**Reviewer #2 (Ralf Loritz)**

Comment:

I do agree with the authors that the discussion about the different modelling philosophies is sometimes rather driven by emotions than by facts. I also think that an opinion on this issue and proposals for a way forward could be of interest for publication in HESS. However, I believe that before this paper can be accepted for publication substantial revisions are needed.

*Reply:*

*We highly appreciate the reviewer's positive, open and highly constructive comments. He raised quite a few points that made us reflect more about the actually underlying issues and we will incorporate his comments as fully as possible in the revised version of the manuscript.*

Comment:

First of all, both authors have a separate opinion paper or comment with a closely related content in HESSD at the moment (Clark et al., 2017; Savenije and Hrachowitz, 2016). Especially the discussion and the review of the opinion paper by Savenije and Hrachowitz (2016) cover a lot of similar points and arguments as this paper. But also the comment by Clark et al. (2017) has several overlapping arguments, especially related to the proposal about how to progress in hydrological modelling. With three papers in HESSD covering similar topics I think it is especially important that the authors clearly show what this opinion paper differentiates it from the other two manuscripts.

*Reply:*

*The reviewer is right in pointing out that we have separate opinion papers, either in review or recently published in HESS. That was not planned and as often in life, things frequently seem to temporally culminate. As a background information: we, the authors, participated in last year's workshop on "Improving the Theoretical Underpinnings of Hydrologic Models" in Bertinoro, Italy. Among the other three dozen participants were some of the most experienced modellers in the discipline of hydrology. Notwithstanding this high level of expertise, one of the most (and most emotionally) discussed topics during this workshop was the difference between different modelling strategies as well as their respective theoretical/physical basis (and lack thereof). As we found these, and further discussions after the workshop among ourselves, the two authors, highly instructive we believe that sharing the different points of view and offering some sort of synthesis may help to direct future efforts in modelling towards more effective developments.*

*Quite naturally, the resulting manuscript then aimed to communicate our opinion on how we, as community, need to understand and approach the modelling problem, which touches the core expertise of both of us, and is thus somewhat related to our respective ongoing work. Notwithstanding the same general topic, i.e. the state-of-art and future needs for modelling, we actually do not see too much overlap between the mentioned papers. Here, our main objective is to resolve the perceived dichotomy between different modelling strategies, which is, in our opinion exactly and exclusively that: perceived. In other words, we intended to make the point that all models*

*are fundamentally the same and that they mostly only differ in their degree of resolution (i.e. complexity): what is the spatial resolution of the model domain (spatial complexity)? Similarly, to which degree do we resolve or lump different processes in our representations (process complexity; see example in S2)?*

*In contrast, on the one hand the Savenije and Hrachowitz (2017) paper emphasizes the need to account for system characteristics that evolve over several spatial and temporal scales if we want to improve our understanding of the hydrological system but also our predictions. On the other hand, Clark et al. (2017) provide a general discussion of (amongst others) the need for a better understanding of scaling in hydrological systems, without making the direct link to top-down/bottom-up models.*

*In this sense, we believe that our manuscript provides additional value by providing a synthesis and suggesting a more stream-lined approach to modelling, arguing that the actual challenges lie in identifying parameters at the relevant scales and which equally apply to both (perceived) endpoints of the modelling spectrum.*

*In any case, we will provide a clearer distinction between the mentioned papers and provide a clearer positioning of this manuscript in the context of existing literature.*

Comment:

My second concern is that a substantial part of this paper reads like a text book. While the language is clear and easy to follow, I was wondering if the potential audience really needs a two page long introduction to "conceptual" and "physically-based" models? Similarly, other sections seem to be redundant as they have already been covered in great detail in several opinion, comment and review papers (e.g. Bahremand, 2015; Clark et al., 2015; Gupta et al., 2012).

*Reply:*

*We acknowledge this point raised by the reviewer. The reason we included a short background on different models was that we think many of the discussions around the use of a specific modelling strategy arise from miscommunications and misunderstandings. We agree, that a modeller will understand and interpret his/her model together with its advantages and disadvantages in a meaningful way. However, we also think that any other modeller will do so in a different way. For example, is there a clear understanding in the community that conceptual models originate from lumped unit hydrograph approaches and that what they essentially do is reproducing observed dispersion characteristics in a signal processing sense and that they can be implemented at any level of complexity (see above), which comes along with the need to converge towards physically based models? We do not think so. Otherwise it would be surprising why many modellers dismiss conceptual models per se as having no physical basis (which may be true for specific implementations, though). To avoid these types of misunderstanding, we believe that in a paper that intends to provide a synthesis of the situation, at very first common ground needs to be established to avoid further misunderstandings. Therefore we also think that a short description of different approaches needs to be part of this manuscript. However, we will change the section to provide an actual framework for a more rigorous model taxonomy.*

*We agree, that some points discussed in our paper have already been covered previously. However, in most cases only individual aspects were discussed. While for example, Bahremand (2016) emphasises the need for parameter allocation to replace calibration, Clark et al. (2016) put the focus on the value of synthesis of hydrological understanding for developing testable model hypotheses and the associated need for more rigorous model evaluation. In contrast, the main intention here is to develop and communicate the point that all model types have, if well implemented, a robust physical basis, albeit at different scales, and that they essentially share the same problems (e.g. need for calibration, hypotheses that are difficult to test with available data, etc.). We will make this clearer in the revised manuscript.*

Comment:

This brings me to a more general comment aimed at all opinion papers which is that careful reading is required to identify where facts end and the opinion of the authors starts. One example for this paper is when the authors write that top-down models have "a parsimonious representation of the energy balance". Is this a fact and has it been shown somewhere or is this an opinion? As far as I know, most hydrological models do not close the energy balance or even keep track of the energy in the system. How can you know if you close the energy balance, when you only try to close the mass balance?

*Reply:*

*This is an interesting point, which we are glad to clarify. We fully agree that most models do not track energy through the system in a detailed way due to the complexity of the processes involved and the lack of data to meaningfully constrain/test potential model formulations of these processes. However, this does not mean that energy is not considered.*

*Energy input is a first order control on the partitioning of water fluxes into drainage and evaporation/transpiration. This partitioning is (or better: needs to be) present in any model. Posing that potential evaporation is a meaningful proxy for incoming energy, the modelled actual evaporation/transpiration then approximately closes the energy balance if: (1) the modelled partitioning between drainage and evaporation/transpiration reflects the observed partitioning (i.e. runoff coefficient) and (2) there is negligible inter-catchment groundwater flow. While the latter point is arguably difficult to test in most catchments, a model can be constrained to reproduce a meaningful partitioning pattern by not exclusively calibrating it to stream flow, but simultaneously also to the runoff coefficient (e.g. long-term average, inter-annual and/or seasonal). This is quite evidently a simple black-box approach to the energy transfer in hydrological systems but it allows the system overall energy balance to be approximately closed. In other words, the energy balance is implicitly and in a simplified way present in the runoff coefficient (see also the Budyko relationship). From that perspective, if well implemented (as stressed on page 6,l.24-27 in the original manuscript), top-down models do at least not considerably violate the energy balance. Even if this is not explicit, we do not think that this is an opinion but rather a physical necessity. We will further clarify this in the manuscript.*

Comment:

Another example is the unclear separation of the macro- and microscale in this paper. For instance the authors argue that macroscale models are important and physically-based with e.g. Sivapalan's (2005) search for a general law at the macroscale or with a comparison with Gay Lussac's law. However, the papers they mention to support this argument use often macroscale models to define various states of the microscale, for example the root zone storage. While using macroscale models to estimate states at the microscale is a perfectly valid approach, it is very important to make clear to the reader that this can only be an estimate and is rather difficult because of the high degrees of freedom we have in hydrology. A precise definition of the macro- and microscale and a clear structure of the manuscript in this context might help to improve this paper and would ensure that not even more "modelling myths" are generated.

*Reply:*

*This is a very good point and we fully agree with the reviewer. We will provide a clearer definition of macro- and microscales. We understand the microscale as the scale at which direct observations of the system boundary conditions/parameters are typically available, i.e. soil sample, plot scale, individual plants, etc. These values emerge from yet smaller scale processes and heterogeneities at the scale of the actual observation and are fully valid for the domain they have been determined for. However, and quite obviously, they cannot by themselves account for spatial heterogeneities between the sampling points and the feedback effects arising from these. For example, interception capacity can be determined for some individual plants (or groups of plants; i.e. microscale) but is problematic to transfer to other parts of a system or to scale-up as it is influenced by a range of different factors, including but not limited to plant species, plant age, plant shape, plant location (wind exposure!), composition of different plant species, or spatial densities of individual plants.*

*The macroscale is then, in our understanding, the scale at which the heterogeneities of the individual microscales observations and in-between are integrated to emerge as functional relationships (see example S2). These are not directly observable using standard observation technology. Yet, there is potential for quite robustly inferring at least some of them through analysis of domain (e.g. catchment) scale data (e.g. runoff). Some examples include the time scale of the groundwater or the root zone storage capacities.*

*The reviewer is correct in assuming that macroscale values are then estimations and it is, as they represent the domain integrated picture, difficult to infer spatial patterns beyond the domain they have been developed for (e.g. catchment, HRU, grid cell, etc.). In other words, when using a lumped model with a lumped root zone storage capacity, this capacity will very well represent the system overall capacity, but intra-domain spatial differences cannot not be readily extracted.  On the other hand, basing the root zone storage capacity on microscale values, i.e. point observations in a distributed, bottom-up model formulation will allow a representation of spatial pattern. This, however, comes at the price that the spatial heterogeneities between the (typically scarce) observation points and therefore the system overall capacities are likely to be not well captured, thus introducing uncertainty.*

*In this context we politely disagree with the reviewer, because we do not think that a macroscale parametrization does result in more degrees of freedom than a microscale parametrization. It is true that at this point most macroscale parameters cannot be observed and tested against data and therefore require calibration (with all its adverse effects). But the same is true for microscale parametrizations: they are strictly valid for their points of observations but not beyond that. As we do not have spatially seamless observations, using these parameters to describe the entire domain either provides a false sense of model accuracy or requires additional calibration (for a much higher degree of freedom than the macroscale representation). Thus, to be cheeky: there is no free lunch, or both ways currently still have substantial drawbacks.*

Comment:

However, I believe that we do not need another paper where we discuss how physically-based or not the different modelling philosophies are. I recommend that you focus on the complementary merits of both approaches. Furthermore, I suggest giving clear examples and sharing your ideas how we could for instance combine top-down and bottom-up models in practice. This could make the manuscript much more unique and meaningful. As I believe that the discussion of this topic is of relevance for the hydrological community I hope my comments, questions and opinions are constructive and can help to improve this manuscript.

*Reply:*

*We agree with the reviewer and we will give more emphasis on the complementary merits of the approaches and how these can best be exploited. We think that a fruitful way forward will be let spatial and process complexities of top-down models converge towards the representations in bottom-up models and vice-versa, with the overall aims of formulating models that can better satisfy the contrasting priorities of (1) a meaningful representation of in particular spatial patterns and (2) meaningful tests of the underlying hypotheses while (3) keeping the required degrees of freedom at a minimum level.*

Comment:

Page 2 Line 28: Maybe add some references where the authors showed that their model failed after the calibration period, both from the bottom up and top down community.

*Reply:*

*Agreed, we will provide some references.*

Comment:

Page 3 Line 6: Could you define catchment scale?

*Reply:*

*We define catchment scale here as all scales starting from a few hillslopes that are drained by a 1st-order stream.*

Comment:

Page 3 Line 10-11: What do you mean here with "respect to bottom up models".

*Reply:*

*We will re-formulate this sentence.*

Comment:

Page 3 Line 11 – 12: I couldn't find the part where you provide a perspective of how to take advantage of different modelling philosophies.

*Reply:*

*We will clarify that and put more emphasis on this aspect in the revised manuscript.*

Comment:

Section 2 Modelling philosophy: This section is mostly written clearly and precisely. Nevertheless, I think the potential reader of this opinion paper is already familiar with the different modelling approaches and reading this section is very akin to reading a text book. I would consider shortening this section with references to other studies or textbooks.

*Reply:*

*We fully agree that the reader will be familiar with the different approaches, but not necessarily with their origin/background. This and common misunderstandings do in our opinion call for the need to establish common ground (see reply to comment above). However, instead of providing the general backgrounds of the two model approaches, we will give a wider view and provide a more systematic framework for an actual model taxanomy.*

Comment:

Page 4 Line 4 – 5: From my point of view the scenario in which you end up in a catchment where you only have reliable runoff and rainfall data but nothing more available is rather unrealistic: In which catchment in the world do you have reliable streamflow, evapotranspiration and rainfall measurement but no other information of the catchment? At least in Europe and the US you have land cover and geological maps. Furthermore, if there is a gauging station and rainfall measurements, most likely a person is doing maintenance on the respective instruments on a regular basis. This

person will most likely accumulate a lot of qualitative information about the hydrological functioning of the catchment and could possibly also complement this picture with low-effort additional measurements or soil sampling. For instance Jackisch et al. (2014) showed how fast one can characterize a remote meso-scale catchment based on a brief measurement campaign. If land cover is managed forest or agriculture, frequently nationwide reports on productivity and for example drought risks are available. We have digital elevation models for the whole earth in decent resolution, monthly estimates of precipitation and soil moisture from satellites and so on. In my opinion the problem is often very different from the projected scenario: We do not know how to use the data in our hydrological models or if it is of relevance. But I admit that this may be a different story.

*Reply:*

*We fully agree with the reviewer and did not intend to imply otherwise. What was meant here is hydrometeorological data. As at this point no generally valid, reliable and quantifiable functional relationships between factors such as topography, geology or soil types on the hydrological response are available, these data are very valuable to develop and constrain models but cannot be a stand-alone replacement of actual hydrometeorological and hydrological observations (i.e. precipitation, stream flow, etc.). In the absence of detailed, spatial high-resolution observations of fluxes and states (cf. the boundary flux problem), conceptual models are therefore mostly developed on the basis of what is available, which is in most cases stream flow, precipitation and estimates of potential evaporation. Of course we agree with the reviewer that the mentioned system characteristics should and eventually need to be used to develop meaningful models, a point which we explicitly address in section 3.2 and its subsections.*

*Similarly, we fully agree that we need to be more efficient in extracting information from our data, which boils down to the paragraph starting at page 8, line 27 in the original manuscript: "The lack of an adequate model calibration, testing and evaluation culture partly arises both from insufficient exploitation of the information content of the available data, and also the real lack of suitable data to more effectively constrain models [...]"*

Comment:

Page 4 Line 5-8: Is the "system integrated response pattern" really the "starting point" of top-down models? Isn't the starting point the delineation of a catchment based on the surface topography assuming a closed water balance? Since most top-down models are calibrated on the streamflow, do you mean streamflow by the term "system integrated response pattern"? Consider clarifying what you mean with the terms, maybe some examples beyond stream flow, and what you mean with "starting point" here.

*Reply:*

*By system integrated response pattern we mean time series of stream flow and other hydrological signatures that can be constructed from these time series (e.g. flow duration curve), or similar variables that characterize the overall flow domain, e.g. solute concentrations in the stream. These system integrated observations are in contrast to point observations of system states, such as*

*groundwater levels or in-situ soil moisture observations (we do on purpose not mention remote sensing products that claim to provide soil moisture estimates, as it is not clear what these different products actually indirectly estimate and how this information can best be used in models).*

*All models need to be based on a definition of the flow domain, i.e. estimates of contributing area or catchment area, and on conservation of mass. This is, however, where the two modelling approaches start to diverge. In this sense, the starting point of top-down approaches is the system integrated data, such as stream flow, when defining the problem as: "We have observations of precipitation input signals and observations of how these input signals are dispersed as stream flow – what is the associated low-pass filter (i.e. model formulation)?". In contrast, the theoretical starting point for bottom-up models is the detailed knowledge of the flow domain and its processes, from which the system integrated response pattern (should) emerge.*

*We will clarify this in the revised manuscript.*

Comment:

Page 5 Line 12: Could you please explain in more detail what you mean with a parsimonious representation of the energy balance?

*Reply:*

*Please see reply to associated comment further above.*

Comment:

Section 3 Modelling myths (C1) "Top-down models have a poor physical and theoretical basis": Comparison with Gay-Lussac's law: I think that the comparison with Gay-Lussac's law and the top-down modelling approach is a little misleading. I am not saying top-down models are not physically based. Like most hydrologists I believe that this entire discussion is based on an ill-posed definition and classification of hydrological models into the dichotomy of physically-based and conceptual models. However, with Gay-Lussac's law you can describe the macroscopic state of a system. But you can't say anything about the microscopic state of the system, for example where the molecules really are. Following your arguments and speaking of topdown models now this would mean that you can't say anything about the microscale of a catchment, for example where the water is in your catchment. However, later you argue that you can identify the root zone storage with a top-down model. Is this not part of the microscale? With a macroscopic model you can only infer about the microscale if you constrain the possibilities of the microscale using either additional measurements or process-based reasoning with the help of statistics. However, this is really difficult in hydrology due to the large number of degrees of freedom. For example, if your model is calibrated to mimic the runoff generation and if we assume for a second that the two water worlds proposed by McDonnell (2014) are real, there is no information about the root zone storage in the rainfall-runoff data and it is really difficult to know if what you learn from your models is true.

Overall, it is not clear where you want to go here. A top own model is based on 1.) the conservation of mass and 2.) on the delineation of the landscape into some kind of control volumes mostly in form of a catchment. With a top down model you can hence make assumptions about the macrostate of a catchment or of a similar control volume. With the help of statistics, process-based understanding or additional measurements you might be able to get a grasp of the microscale. So why are you comparing it with a natural law which is constrained by the energy and mass conservation when the model you defend is not? I believe most hydrologists know how a conceptual model works so is this whole comparison necessary at all? Maybe a rigorous definition of macroscale and microscale might help to improve and clarify differences, similarities and linkages between top-down and bottom-up models?

*Reply:*

*We agree with the reviewer that the two systems (gas volume vs. catchment) do have structurally different characteristics. For this reason we understood the comparison as "analogy" (according to the Oxford Dictionary: "partial similarity") and not as full "similarity". We further agree with the reviewer that the microstates are unknown in the gas volume. Similarly, we believe that in principle microstates are, to a certain degree (dependent on the available data and the chosen model resolution/complexity) unknown in a hydrological model (model macrostates rather "integrate […] natural heterogeneity within the model domain […]", p.13,l.33 and elsewhere in the manuscript). A fully lumped, one-process model (e.g. one bucket with a non-linear storage-discharge relationship) would come very close to the conceptualization of a gas-volume. A model that accounts for more individual processes and higher spatial resolution will move away from that situation. Thus, we think that given the roots of top-down models (e.g. unit hydrograph) the analogy is not too far-fetched. However, we all know that such simple models do not do a good job in representing real world heterogeneity in hydrological systems. The inherent difference between the gas volume (or one-box systems) and catchments (or more detailed models thereof) is that a purely statistical (or data-driven) approach is, following the argument of Dooge (1986), only applicable for systems in the realm of unorganized complexity (i.e. high degree of randomness and complexity). While catchments are too random and complex for an exclusively mechanistic treatment, they are equally not random and complex enough for an exclusively statistical treatment – they rather fall into the realm of organized complexity. In other words, some structure, i.e. distinction of individual processes and/or spatial discretization, is required to meaningfully represent the system. However, within this structure (i.e. within the individual model components, such as the root zone, or, if spatially discretizing, within a given e.g. landuse class) the same principle applies: relatively stable relationships that integrate the sub-domain heterogeneity emerge at the scale of the model domain. In this respect (and for the sake of the argument, assuming no spatial discretization), it is true that the microstates of the root zone cannot be identified and it is not known \*where\* in a catchment how much water is stored in the root zone. Rather, what is known is that water is stored in the root-zone component and not in e.g. the groundwater component of the system. Therefore, we do not think that the points mentioned by the reviewer point towards a contradiction. Of course the argument can be extended in the same way, when adding spatial discretization, e.g. into land use classes. Each land use class will then be represented by emergent relationships that integrate the sub-domain heterogeneity of this very class – again reflecting the basic idea of knowledge of macrostates without the knowledge of microstates. We will, due to the reorganization of the manuscript, remove the gas laws analogy and provide a clarification for the importance of organization in the revised manuscript.*

Comment:

Page 6 Line 3: The molecular dynamics approach might be untestable and unfeasible but certainly not unnecessary. It is the theoretical basis of the movement of gas particles and hence necessary if you want to understand a system.

*Reply:*

*Agreed, we will reformulate that statement.*

Comment:

Page 6 Line 23: Can you please explain in more detail what you mean with parsimonious representation of the energy balance, again?

*Reply:*

*Please see reply to associated comment further above.*

Comment:

Page 8 Line 1: Holistic empiricism and on Page 7 Line 6 assign physical meaning to them a priori? Please explain why the two statements are not in contradiction.

*Reply:*

*We think there is some value if the system was seen from the perspective of holistic empiricism – not necessarily that the complete system has to be treated as fully holistic. Assigning parameters derived from observations at the modelling scale and thus assigning physical meaning to individual model components does not contradict the holistic perspective: the parameters obtained from observations at that scale fully integrate the system-internal heterogeneity and its internal interactions. This therefore directly links to holism, which poses that in an interconnected system only sets of hypotheses (i.e. processes at the scale of observations) but not individual hypotheses (i.e. processes at the sub-grid scale) can be meaningfully tested.*

*The point that distinguishes hydrological systems here is that they are mostly in the realm of organized (i.e. structured) complexity (Dooge, 1986). In other words, they are systems that are characterized by clearly distinct "groups" of processes or components. While it is difficult to reconcile all these components in a truly holistic hypotheses, we believe that each individual component may well be described using the holistic perspective. Thus, we agree, in principle with the reviewer that these two statements are contradicting each other. However, we would argue that this is not the case if the organized nature of catchments is brought into consideration.*

Comment:

(C2) "Top down models are too simplistic…" and (C3) "Top-down models are ad-hoc formulations…": Both sections are written clearly and well but I think this has all been said and written down several times. You might consider to shorten this section.

*Reply:*

*This may be true. Yet, the same discussion is coming up over and over again: "Top-down models are too simplistic…" or "Top-down models are ad-hoc formulations…". Therefore we think there is some value in bringing together the loose ends here by analysing the question from both perspectives (which we will try to improve in the revised manuscript)*

Comment:

Subsubsection 3.2.2 and 3.2.1: What do you mean with process and spatial complexity. Could you please define complexity and how it relates to the respective models?

*Reply:*

*We use the term "complexity" here to refer to "resolution". Process complexity thus describes, how many individual, interacting processes are considered to generate the response (see example S1). Spatial complexity describes the spatial resolution of the model domain. We will clarify that in the revised manuscript.*

Comment:

Page 10 Line 30-31: Is it really multivariate observed response dynamic? At least in one of the cited examples the authors only use streamflow and derivations of it.

*Reply:*

*Good point. What was meant is "multi-objective", which may include both, multiple variables and multiple objective functions. We will reformulate that to be more precise.*

Comment:

Page 12 Line 19 - 20: "Competing approaches" Despite the title of the manuscript I had the feeling that the main focus was on defending top-down models. Why do you stress the dichotomy although I understood the overall aim of your opinion paper to be exactly the opposite?

*Reply:*

*Agreed. As mentioned in the reply to one of the comments further above, we now realize that the paper comes across as a simple defence of top-down models. That was not our intention. We tried to resolve the perceived dichotomy between the two model approaches ("all models are physical, all models are conceptual") and we will put some effort to do so in a clearer and more obvious way.*

Comment:

Page 13/14 Line 34 / 1-2: I think this sentence is a little misleading. Obviously you can use a DarcyRichards based model on the macroscale. However, you need to use a rather fine discretization of the model elements.

*Reply:*

*Fully agreed, we will reformulate this statement to be more precise.*

Comment:

Page 14 Line 16 - 17: Why are you so pessimistic here? Maybe you could add some references so the reader can better understand your pessimism.

*Reply:*

*This is a pointed statement, which was qualified by the stating "in an exaggerated way" (p.14,l.16). However, we think that there is some truth to it, without being pessimistic. We will substantiate the statement with references to work on data/parameter uncertainty (e.g. Beven, Westerberg, McMillan) and uncertainties arising from the model building process (e.g. Gupta, Clark, Wagener).*

**Reviewer #3 (Thorsten Wagener)**

Comment:

The authors, as always in their papers, have written a well-formulated discussion of relevant current issues in hydrological modeling. While there are many interesting points here, and Hoshin points out quite a few, I have to agree with Ralf Loritz's comment that it becomes hard to keep track of what they key points are in an increasing number of commentaries on (at least seemingly) similar issues. In the case of the present manuscript, I think that there are some issues that can be discussed with more rigour to highlight its uniqueness (though the authors might disagree).

*Reply:*

*We highly appreciate the reviewer's positive assessment of our manuscript. We also agree that the argument needs to be sharpened to more strongly emphasise our intention and main message, which we will try to do in the revised manuscript.*

Comment:

One thing that stands out in this commentary is the explicit use of the term top-down modeling. It is not clear to me though what definition the authors use for top-down modeling. My understanding of the manuscript suggests that here this definition includes all conceptual type approaches to hydrologic modeling. So, are all conceptual modeling approaches equivalent to a top-down modeling philosophy? I do not think so, though the authors likely have a different point of view (which would be fine). What definition do the authors follow? Is this defined by the model type I use (ODE vs PDE) or by the mindset/objective I have when developing my model?

Following some of the early definition top-down modeling "provides a systematic framework to learning from data, including the testing of hypotheses at every step of analysis" (Sivapalan et al., 2003). This is often applied in a hierarchical manner (e.g. using signatures), but not necessarily so. If this is the definition the authors use, then I do not think that models such as HBV have been developed following a top-down modeling philosophy. They rather have been developed with a bottom-up mindset I think. Similarly the Sacramento model was not build to just fit the data, but based on an attempt to provide a simple representation of physics. Is there really a common philosophy underlying the modeling approaches used to build HBV, in the top-down papers by Sivapalan and colleagues, and in the FUSE framework? Is it really a binary decision whether an approach is top-down or bottom-up?

*Reply:*

*We indeed started from the premise that top-down modelling "provides a systematic framework to learning from data, including the testing of hypotheses at every step of analysis". However, we realized that the distinction we used in the original manuscript was not precise enough. In fact, we think that much of the misunderstandings between different modelling approaches originate from the fact that terms such as top down, conceptual, bucket, lumped on the one hand and bottom-up, physical and distributed on the other hand are often used interchangeably in spite of having only*

*limited overlap. We will therefore widen the scope of the paper here and provide a framework for a more systematic model taxonomy. This framework will allow to place all model approaches on different positions in the spatial resolution-process complexity spectrum between the two endpoints and will highlight the fact that all models are physical and to some degree conceptual.*

Comment:

If I assume that the definition by Sivapalan above is appropriate, then some important contributions to top-down modeling are missing from this paper. Most notably is the work by Peter Young (e.g. Young, 2003 and much earlier than that), who, with his databased mechanistic approach, has provided one of the few very structured frameworks for top-down modeling. Of course he did so by making some strong assumptions, which limit the generality of his approach. It would be good if the authors could have a wider look at literature in which top-down modeling strategies are investigated (if they use the term more narrowly than simply all conceptual models).

*Reply:*

*We thank the reviewer for this very good point – of course the databased mechanistic approach needs to be part of such a discussion. We will add relevant references in the revised manuscript.*

Comment:

I think by using a very wide definition of top-down modeling, we miss the opportunity to discuss some important remaining problems. Mainly that hydrology still lacks "a systematic approach to learning from data" as proposed by Siva. For example, how do we assess model complexity (given that information criteria typically do not work for hydrologic models), so that we can identify the simplest model that fits the data? How do we decide that one model structure is better than another one beyond just looking at performance? The data-based mechanistic approach provides a nice strategy to identify the simplest representation (of routing) supported by the data, while also allowing for a hydrological interpretation. I do not think that we have a more general framework of this type yet (i.e. without Peter's assumption of using linear transfer functions etc.).

*Reply:*

*We agree that these are important points (which are raised in section 3.2) and we will add these aspects and related papers (such as the work of Patrick Willems) to the relevant discussion.*

Comment:

I am also unclear why a top-down approach should be restricted to catchment scale observations (if that is what the authors suggest). If the approach is focused on learning from data then its philosophy can be applied at any scale. Work by Young and colleagues using their top-down philosophy have not been limited to catchment scale hydrologic data, so why should it be for us in

hydrology? We could actually build distributed models using a top-down strategy for catchments with extensive internal observations.

*Reply:*

*We fully agree and think that we did not imply otherwise. That is why, throughout the manuscript we tried to speak in terms of "model domain", which can be e.g. a catchment, a HRU or a grid cell. However, as many conceptual models are formulated as lumped representations, we explicitly referred to as catchments as the model domain in these cases. But quite clearly, the top-down approach in its essence is applicable, and should be applied, at any scale, which would then in term the convergence towards detailed bottom-up models, which we think is necessary. We will clarify that in the revised manuscript.*

**Reviewer #4 (Marc Bierkens)**

Comment:

I started to read this opinion paper with great anticipation because I think there is a desperate need for joining top-down and bottom-up approaches to arrive at solid hydrological theories. The paper is generally well written and starts out with a promising small review about the nature of bottom-up and top-down approaches.

*Reply:*

*We highly appreciate this positive assessment.*

Comment:

However, after reading the part thereafter, I have to admit I started to become a bit disappointed. The reason for this is that the second part of the paper becomes quite unbalanced and reads as an apologia for top-down modelling. What I miss is a section "Modelling myths or not" for bottom-up approaches. For example, statements as "Bottom up models are over-parameterized" can be elaborated on. After that I would have liked to have a section to sketch a way forward to marry both approaches taking account of their complementarities. Shortening the "Modelling myths or not" to make room for similar sections on bottom-up approaches would make the paper much more balanced and interesting.

*Reply:*

*Reflecting also the points raised by Reviewer #1 and after re-analyzing the manuscript from the perspective of all reviewers, we fully agree that it comes across more like a defence of top-down models rather than the intended balanced evaluation of the two modelling strategies. We will accordingly re-structure and re-formulate the relevant sections in the revision by adding perspectives towards the bottom-up approach. We will also put more emphasis on how to take the best out of both approaches.*

*Having said that, and given that also the other reviewers noted that the paper should be less a defence of top-down models, we would also like to stress one, potentially not irrelevant point: bottom-up models, i.e. "physically-based" may largely benefit from a semantic-psychological bias. The term "physical-based" inherently implies that they are "correct" descriptions of real world-systems, which further implies that all other models are not "physical" and thus less "correct". From this perspective, we believe that any type of comparison between bottom-up and top-down strategies will to some extend necessarily come across as a defence of top-down models, i.e. explanations of why they can be as meaningful representations of reality as bottom-up models. In other words, already the term "physically-based" puts bottom-up models in the (often not really justified) position of benchmarks other models have to be compared to, even if they are not necessarily "better" descriptions of reality.*

Comment:

First, the authors underpin the statement that "At the macroscale, which in the realm of organized complexity is frequently characterized by the emergence of relatively simple functional relationships. . . that integrate typically unobservable natural heterogeneity over the model domain", with a comparison with to statistical physics (e.g. gas laws). However, there is a big difference between an ideal gas and a hydrological system related to the assumption of ergodicity. In that context, this assumption loosely means that at all times all microstates are present when averaging over the volume. This assumption is valid for an ideal gas but not necessarily the case for hydrologic systems.

*Reply:*

*We fully agree with the reviewer that there is no full correspondence between the two systems. We rather understand it as an analogy, i.e. a partial similarity, of the systems. In our understanding, the essential difference between the two systems is that a volume of an ideal gas is random and complex for it to be considered in the realm of unorganized complexity, where the microstates of large enough samples can be meaningfully characterized on the macroscale based on their statistical properties (ergodic system). In contrast, hydrologic systems are characterized by lower degrees of randomness and complexity and thereby fall in the realm of organized complexity, where systems cannot be fully described by statistics alone. We believe, that organisation in catchments is manifest in the structure of the hydrological response, which in turn is caused by the varying connectivity of processes acting on distinct time scales. In other words, depending on the wetness history and the "memory" of the system, any combination of these (statistically different) processes can be active at a given time. In spite of this overall structure (or organization), we think that the at least some of the individual processes may well approach the definition of ergodic processes (of course given the full knowledge of input, output and boundary conditions). An example may be the groundwater dynamics at the catchment scale: during low flow periods, the drainage of the "deep" ground water is the only processes sustaining stream flow (and due to the depth of the groundwater table only negligible evaporation is occurring) in many catchments world-wide. At the catchment-scale this emerges as exponential recession characteristics and thus suggests simple linear storage-discharge relationships. We think that it is not unreasonable to assume that a random samples of this process will reflect the statistical moments of the full process, which is the fundamental definition of an ergodic process. However, due to the reorganization of the manuscript we will remove the gas law analogy in the revised manuscript.*

Comment:

Second, I feel that a problem with the way top-down megascopic hydrological laws are derived (also in comparative hydrology) is that often only (signatures) of the output variables are used to assess the form of the Q = F(S) relationship. This can only be done if a certain form (often a power function) is assumed a priori. I think that to really assess the form of these relationships one needs to jointly measure the state (groundwater storage, soil moisture, snow water equivalent) and the output variables (discharge, evaporation). Very rarely these state observations are used or available in

catchments used in comparative hydrology. So we should get away from the fixation with hydrographs only and start measuring states. To add to this: energy conservation is often added by checking if the found megascopic laws follow Budyko's hypothesis. This is only a weak check on energy conservation, because it only checks for very long times and doesn't guarantee energy conservation at any given time.

*Reply:*

*We also wholeheartedly agree with this comment and do not state otherwise in the manuscript. We would also take this point a step further and argue that the problem does not only apply to top-down models. Bottom-up models, based on extrapolations of anecdotal observations, are not unlikely to suffer from similar problems. Remote sensing products do have the potential to allow for real progress here (e.g. GRACE). Another point that is currently not fully exploited is the information content in spatial patterns. We think that systematically forcing (semi-)distributed models to produce good correlations with observed spatial pattern of, for example, soil moisture or snow cover will prove highly valuable to test models.*

*We also agree that the Budyko framework provides a models test, albeit a very weak one. However, the actual observed runoff coefficient may hold more information, as it cannot only be applied over the long-term, but models can also be trained to reproduce annual or even seasonal sequences of observed runoff coefficients. Doing this will strengthen the test on energy conservation (albeit not fully solving the problem, of course). We will clarify this in the manuscript.*

Comment:

Third, once megascopic laws have been derived empirically, these laws' physical basis should be strengthened by also deriving them from upscaling from smaller-scale mechanics. A well-known example is Darcy's law. It was first established empirically - note that this was done by both observing states (heads or actually the head gradient) and fluxes. Later (much later), it was shown that it could be derived from the Navier-Stokes equations (by 1. neglecting quadratic inertia terms: laminar flow -> Stokes equations; 2. volume averaging by homogenization; 3. noting that drag forces are much larger than viscous forces). Obviously, heterogeneities in hillslopes and catchments are more complex than pore-scale heterogeneities in a REV. This makes simple homogenization not likely a suitable approach. However, hyper-resolution (cm-scale) modelling using simulated heterogeneities (including macropres etc) with 3D PDE-based models (e.g. Parflow, Hydrogeopshere, Cathy) and upscaling the results may be a way to derive megascopic laws from first principles.

*Reply:*

*We agree with this. That was also the motivation behind the statement: "While top-down models approach the problem from a macroscale physical understanding, bottom-up models emphasize the microscale perspective. An ideal model would, almost needless to say, provide an equally good representation of both aspects." (p.13,l.19-21). We will clarify this make it more explicit in the revised manuscript.*

**Short Comment (Sivarajah Mylevaganam)**

Comment:

The current version of the paper does not convince that the cited papers are sufficient and informative for the authors to draw conclusions or comments on the topic that is discussed in this paper. Moreover, from the reader's point of view, what has been discussed in this paper has already been echoed in the current literature.

*Reply:*

*We agree, that the overall topic was already subject in earlier papers. However, we feel that the due to a lack of synthesis between these earlier papers, there are still quite some misunderstandings and miscommunications about the background and nature of different modelling approaches within the hydrological community. We will clarify this in the revised manuscript and add more relevant references to better support our arguments.*

Comment:

It has been extensively argued in numerous journal papers about the pros and cons of topdown and bottom-up approach. Therefore, from the reader's point of view, for this commentary to have some merits, the authors need to go beyond what has been understood in the current literature. From the reader's point of view, it would be more useful, for example, if the authors bring the concept of middleware that lies in between the said approaches of modeling (i.e., top-down and bottom-up).

*Reply:*

*We agree and that is exactly the intention of this manuscript: different model approaches need to converge for further progress in the discipline. We will make this clearer in the revised manuscript.*

Comment:

In the current version of the paper, the authors scrutinize common modelling critiques (C1-C3). Are these critiques developed by the authors? Are these critiques developed based on some published survey? What motivated the authors to consider these critiques as the "common" modelling critiques?

*Reply:*

*These critiques are points that often came up in the authors' discussions with modellers from other research groups during international conferences, workshops and joint projects.*

Comment:

In the current version of the paper, the authors scrutinize common modelling critiques on top-down models (C1-C3) and discuss the extent to which they are justified. From the reader's point of view, the title of the paper does not fit the content of the paper.

Reply:

We fully agree and we will re-structure and re-phrase the relevant sections of the manuscript.

Comment:

Referring to line number 22 on page number one, the authors state that the models frequently fail to reproduce the hydrological response in periods they have not been calibrated for, thereby providing unreliable predictions. From the reader's point of view, this statement needs to be cited.

*Reply:*

*Agreed, we will provide suitable references.*

Comment:

In the current version of the paper, the authors discuss about the spatial complexity, process complexity, and spatial scale. However, referring to line number 22 on page number one, from the reader's point of view, it would be more useful if the authors discuss about the influences of temporal scale and its complexity on the said approaches of modeling (i.e., top-down and bottom-up). Is it scientifically justifiable that the processes that are modeled at a particular temporal scale do not change when the temporal scale changes? In the current literature and the modeling practices, the processes that are modeled are the same regardless of the temporal scale of the simulation.

*Reply:*

*Interesting point and we will consider a discussion of this in the revised manuscript.*

Comment:

Referring to line number ten on page number one, a better understanding bears the potential of identifying the complementary value of the two philosophies for improving "our" models. Are these models developed by the authors? Is this commentary about the models developed by the authors?

*Reply:*

*The term "our" throughout the manuscript refers to the hydrological modelling community and the models developed by the community.*

Comment:

From the reader's point of view, some of the paragraphs are repetitive (e.g., the paragraphs about the activation and deactivation of processes).

*Reply:*

*We will analyse and re-phrase where appropriate.*